# Escaping the Likelihood Trap: Geometric Diversity Optimization for Long-Form Image Captioning

Qingmei Tang [1]   Shuai Hao [2]   Rong Fu [3]   Zirui Mo [4]   Xiang Liu [5]   Jiaxuan Lu [6]   Wenyu Wang [1] [†]

## Abstract

The utility of Vision-Language Models (VLMs) in reasoning and auditing tasks hinges on their ability to exhaustively describe visual scenes. However, current models exhibit a pathology we term the Likelihood Trap: standard alignment objectives, specifically MLE and KL-regularization, drive generation toward generic, high-probability templates, systematically suppressing fine-grained details. To overcome this, we introduce Geo-RL, a framework that shifts the optimization target from mode-seeking per-sample objectives toward diversity-oriented set-level RL. Geo-RL reformulates caption generation as maximizing the volume of a parallelotope in semantic space. By leveraging Determinantal Point Processes (DPPs), we enforce orthogonality among sampled descriptions, ensuring that they span the image's full semantic support. Crucially, we derive a closed-form leave-one-out marginal reward, enabling stable policy optimization. Empirically, Geo-RL escapes the trap, achieving a significant improvement in semantic richness and detail coverage without compromising visual grounding.

## 1. Introduction

Vision-language models (VLMs) have reshaped multimodal understanding, enabling systems to describe, reason about, and converse over visual content. Yet as the field

[1]School of Airspace Science and Engineering, Shandong University, Weihai 264209, China [2]Beijing University of Aeronautics and Astronautics, China [3]The Institute of Collaborative Innovation, University of Macau, China [4]School of Electronic Information, Central South University, China [5]Department of Autonomous Driving, NIO, China [6]Shanghai Artificial Intelligence Laboratory, China. Correspondence to: Qingmei Tang <tqm@mail.sdu.edu.cn>, Jiaxuan Lu <ljxlujiaxuan@gmail.com>, Wenyu Wang <hochi@sdu.edu.cn>.

*Proceedings of the 43$^{rd}$ International Conference on Machine Learning*, Seoul, South Korea. PMLR 306, 2026. Copyright 2026 by the author(s).

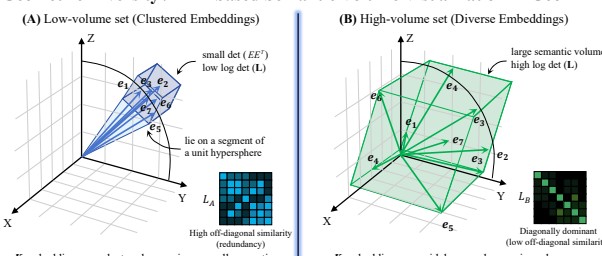

*Figure 1.* Geometric Diversity via DPP-based Semantic Volume in Geo-RL. Given $K$ generated captions, their semantic embeddings define a geometric structure in representation space. Clustered embeddings span a small semantic volume (low DPP log-determinant), indicating redundancy, while diverse embeddings yield a large volume (high log-determinant). Geo-RL maximizes this volume during training to encourage complementary semantic coverage over redundant rephrasings.

shifts from short labels to long-form generation, the central challenge is no longer merely describing an image fluently, but avoiding a single narrow view of a visually rich scene. A capable VLM should produce diverse and faithful descriptions, capturing an image from multiple complementary perspectives. Whether used for generating varied training data or surfacing heterogeneous evidence for auditing, its outputs should collectively span the semantic support of the scene rather than collapsing onto one dominant concept. Under this view, fluency and per-caption completeness are merely prerequisites; the real utility lies in the geometric diversity of generated descriptions, jointly covering foreground objects, subtle attributes, spatial relations, and contextual cues across samples. Recent work further shows that more differentiated semantic alignment improves downstream robustness and reasoning (Wu et al., 2025), indicating that diversity-aware generation is a foundational capability for modern VLMs.

However, a fundamental flaw exists in how current models are trained. State-of-the-art systems, driven by Maximum Likelihood Estimation (MLE) or Reinforcement Learning (RL), suffer from what we term the *Likelihood Trap*. Because models are rewarded for predicting the most probable

words, they tend to "play it safe." When presented with a complex image, they gravitate toward the most salient and generic features, *e.g.*, a large sofa in a living room, while systematically ignoring subtler details such as background objects, textures, or lighting conditions.

This phenomenon manifests as *semantic collapse* in representation space: multiple generated captions cluster tightly together in a frozen semantic embedding space, redundantly describing the same dominant concepts. As illustrated in Figure 1 (A), such clustered embeddings span only a small geometric volume, resulting in a low log-determinant under a Determinantal Point Process (DPP) kernel. Safety penalties and KL-regularized alignment objectives further exacerbate this collapse, discouraging the model from exploring rare fine-grained semantic attributes.

Escaping this trap demands a fundamental shift in how we define a high-quality generation target. For a fixed image, the optimal policy should not simply maximize the likelihood of a single caption. Instead, it should generate a set of faithful captions that collectively span the maximum semantic volume of the scene. We argue that diversity in long-form captioning must be geometric rather than lexical. Semantic coverage should be measured by how much volume the set of captions occupies in embedding space, rather than surface-level word variation.

To operationalize this insight, we introduce **Geo-RL** (Geometric-Regularized Reinforcement Learning). Rather than treating sampled captions as independent sentences, Geo-RL models them as a geometric object, *i.e.*, a parallelotope spanned by their semantic embedding vectors. We use Determinantal Point Processes (DPPs) to measure its volume via the log-determinant of a kernel matrix in a frozen embedding space. As shown in Figure 1(B), widely dispersed captions yield a large semantic volume and high reward, whereas redundant captions collapse to a low-volume subspace. Crucially, Geo-RL provides an efficient marginal volume reward that attributes each caption's contribution to the overall volume, enabling stable and scalable per-sample credit assignment. By directly optimizing the geometric objective, Geo-RL encourages complementary coverage of the visual scene, including objects, context, interactions, and style. Directly, Geo-RL serves as a better diverse captioner; indirectly, it provides a useful component for building stronger VLMs.

Our contributions are threefold:

- We formalize long-form caption degradation as the *Likelihood Trap*, where likelihood maximization and KL-regularized alignment favor generic plausibility over semantic fidelity.
- We propose Geo-RL, a framework that aligns training signals with semantic coverage by maximizing the ge-

ometric volume of generated captions.
- We derive an efficient marginal volume reward for per-sample credit assignment and demonstrate that Geo-RL significantly outperforms baselines on long-caption benchmarks by breaking the trade-off between visual faithfulness and semantic richness.

**Conflict of Interest Disclosure.** The authors declare that they have no financial conflicts of interest related to this work.

## 2. Related Work

### 2.1. Long-Form Captioning and Benchmarks

Paragraph-level captioning highlights the need for long, compositional descriptions beyond single-sentence captions (Krause et al., 2017). Localized Narratives add dense grounding signals that expose whether models mention fine-grained entities and relations (Pont-Tuset et al., 2019). Recent multimodal instruction and caption datasets such as ShareGPT4V scale long-form supervision and evaluation, making semantic collapse easier to observe and quantify (Chen et al., 2023). Recent benchmarks and analyses specifically target long-form caption fidelity and completeness in detailed settings, including DeCapBench and its DCScore protocol (Ye et al., 2025), as well as broader evaluations of long-form captioning systems (Lu et al., 2024a; Wei et al., 2025). Complementary metric suites (e.g., CIDEr, CLIPScore, and BERTScore) have also been widely used to diagnose the diversity–faithfulness trade-off in caption generation (Vedantam et al., 2015; Hessel et al., 2021; Zhang et al., 2020).

### 2.2. Alignment Objectives and RL for Generation

Policy-gradient methods optimize non-differentiable sequence-level metrics in captioning (Rennie et al., 2017). Preference-based alignment extends RL to large language models with KL regularization for stability (Christiano et al., 2017; Ziegler et al., 2019; Ouyang et al., 2022b; Bai et al., 2022a), while alternatives such as DPO improve optimization behavior (Rafailov et al., 2023). Related directions include contrastive objectives for image–text alignment and parameter-efficient adaptation methods that avoid full fine-tuning, such as graph-based PEFT for MLLMs (Cheng & Lu, 2025). Prior work on domain adaptation of visual foundation models further highlights the need to preserve fine-grained semantic cues (Lu et al., 2024b). Recent VLMs and multimodal LLMs (e.g., Flamingo, BLIP-2, LLaVA, InternVL, VILA, Cambrian, Qwen-VL, MiniGPT-4, and LLaVA-NeXT variants) demonstrate strong general-purpose visual instruction following (Alayrac et al., 2022; Li et al., 2023a; Liu et al., 2023; Wang et al., 2025; Lin et al., 2023; Tong et al.,

2024; Bai et al., 2025; Zhu et al., 2023; Li et al., 2024b; Liu et al., 2025). However, stronger backbones alone do not guarantee broad semantic coverage for a fixed input. Geo-RL complements these advances by directly optimizing set-level semantic support under standard trust-region alignment constraints (Ouyang et al., 2022a; Bai et al., 2022b; Schulman et al., 2017), targeting the contraction of meaning coverage induced by likelihood and KL regularization.

## 2.3. Diversity Objectives and Log-Determinant Geometry

Prior approaches to diversity promotion include decoding-time heuristics and token-level penalties, such as diverse beam search (Vijayakumar et al., 2016) and unlikelihood training (Welleck et al., 2020), which often increase surface variation without ensuring set-level semantic coverage. Determinantal Point Processes (DPPs) offer a principled framework for diverse subset selection (Kulesza et al., 2012), with log-determinant objectives linking diversity to volume and information gain. Recent work explores diversity at decoding and representation levels, including sampling failure modes (Holtzman et al., 2020b;a), contrastive or CLIP-guided captioning (Dai & Lin, 2017; Zhang et al., 2022; Li et al., 2023b), and set-level measures such as Vendi (Friedman & Dieng, 2023). Building on these representation-level approaches, methods like Llip (Lavoie et al., 2024) explicitly promote caption diversity during contrastive vision-language pretraining. Furthermore, log-determinant objectives have been applied to structured diversity in NLP (Gong et al., 2014; Yang et al., 2023).

## 3. Method

Geo-RL is designed to maximize semantic coverage in long-form image captioning while preserving visual faithfulness. The central idea is to sample a small set of captions for each image and assign rewards to caption sets whose semantic embeddings span a large volume in the embedding space. A log-determinant objective derived from determinantal point processes (DPPs) provides a global volume signal, while a lightweight pairwise repulsion term suppresses near-duplicate captions.

### 3.1. Setup and Notation

Let $I \in \mathcal{I}$ denote an input image drawn from the image space $\mathcal{I}$, and let $y = (y_1, \ldots, y_T) \in \mathcal{Y}$ denote a token sequence of length $T$ in the caption space $\mathcal{Y}$. A captioning vision-language model (VLM) defines a conditional distribution $p_\theta(y \mid I)$ parameterized by learnable weights $\theta$. For each image $I$, Geo-RL draws $K$ independent captions $\mathbf{Y} = \{y_1, \ldots, y_K\} \sim p_\theta(\cdot \mid I)$ and optimizes $\theta$ using set-level rewards computed over the entire caption set $\mathbf{Y}$.

#### 3.1.1. AUTOREGRESSIVE GENERATION

The conditional distribution factorizes autoregressively as:

$$p_\theta(y \mid I) = \prod_{t=1}^{T} p_\theta(y_t \mid y_{<t}, I), \tag{1}$$

where $y_t$ denotes the $t$-th token in the sequence, $y_{<t} = (y_1, \ldots, y_{t-1})$ represents the prefix consisting of all preceding tokens, and each factor $p_\theta(y_t \mid y_{<t}, I)$ specifies the probability of generating token $y_t$ conditioned on both the image $I$ and the partial sequence $y_{<t}$.

To quantify diversity, we operate in a fixed semantic embedding space: a frozen text encoder $\phi$ maps each caption $y_k$ to a unit-norm vector $\mathbf{e}_k = \phi(y_k) \in \mathbb{R}^d$, where $d$ denotes embedding dimensionality.

#### 3.1.2. SEMANTIC VOLUME MAXIMIZATION OBJECTIVE

Geo-RL interprets diversity as geometric volume in the embedding space. Given $K$ captions $\mathbf{Y} = \{y_1, \ldots, y_K\}$ with corresponding embeddings $\{\mathbf{e}_1, \ldots, \mathbf{e}_K\}$, we define the embedding matrix $\mathbf{E} = [\mathbf{e}_1, \ldots, \mathbf{e}_K]^\top \in \mathbb{R}^{K \times d}$, where each row corresponds to a caption embedding. The *semantic volume* is then defined as:

$$\mathcal{V}(\mathbf{Y}) = \sqrt{\det(\mathbf{E}\mathbf{E}^\top)}, \tag{2}$$

where $\mathbf{E}\mathbf{E}^\top \in \mathbb{R}^{K \times K}$ is the Gram matrix of the embeddings, with its $(i, j)$-th entry equal to $\mathbf{e}_i^\top \mathbf{e}_j$. Geometrically, $\mathcal{V}(\mathbf{Y})$ equals the $K$-dimensional volume of the parallelepiped spanned by the embedding vectors. Achieving a large volume requires the embeddings to be approximately linearly independent, thereby encouraging semantic coverage that extends beyond mere surface-form variation.

### 3.2. The Geo-RL Framework

Geo-RL integrates a global volume reward with a local repulsion term and a grounding-aware reward to balance diversity and visual faithfulness. We describe each component in detail below. Figure 2 illustrates the overall architecture of the proposed framework.

#### 3.2.1. DPP-BASED DIVERSITY REWARD

Determinantal point processes (DPPs) are probabilistic models that assign higher probability to diverse subsets through a log-determinant objective (Kulesza et al., 2012). Geo-RL leverages a DPP kernel constructed from semantic embeddings to encourage caption diversity.

**Kernel construction.** For a set of $K$ sampled captions $\mathbf{Y} = \{y_1, \ldots, y_K\}$ with embeddings $\mathbf{e}_k = \phi(y_k)$, we con-

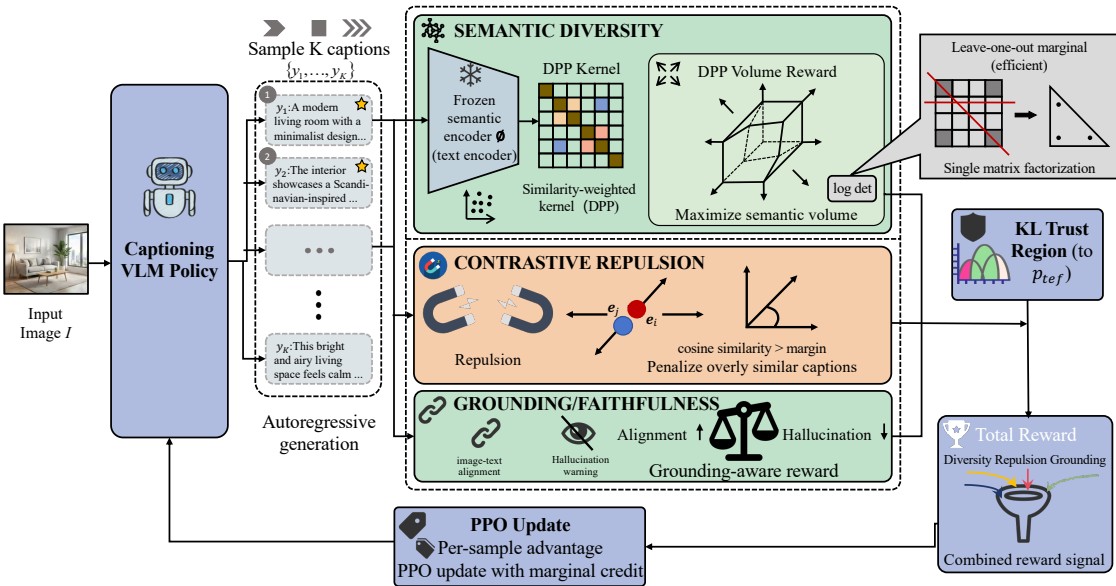

*Figure 2.* Overview of Geo-RL. For each image, the policy samples $K$ captions. A frozen semantic encoder maps captions to embeddings. Geo-RL computes a log-determinant volume reward and a repulsion term from pairwise similarities, and combines them with grounding reward and a KL trust region to update the policy.

struct an L-ensemble kernel matrix $\mathbf{L} \in \mathbb{R}^{K \times K}$ that captures pairwise semantic similarity between captions, with entries defined as:

$$L_{ij} = q_i q_j \cdot k(\mathbf{e}_i, \mathbf{e}_j), \qquad (3)$$

where $q_i \geq 0$ is a quality score for caption $y_i$ (defined below), and $k(\cdot, \cdot)$ is a positive semi-definite kernel function measuring similarity between embeddings. The quality terms $q_i$ down-weight low-quality captions, ensuring that volume maximization is encouraged among plausible candidates rather than arbitrary text.

**RBF kernel.** We employ a radial basis function (RBF) kernel with bandwidth parameter $\tau > 0$:

$$k(\mathbf{e}_i, \mathbf{e}_j) = \exp\left(-\frac{\|\mathbf{e}_i - \mathbf{e}_j\|_2^2}{2\tau^2}\right), \qquad (4)$$

where $\|\mathbf{e}_i - \mathbf{e}_j\|_2^2$ denotes the squared Euclidean distance between embeddings $\mathbf{e}_i$ and $\mathbf{e}_j$. The bandwidth parameter $\tau$ controls the sensitivity to embedding differences: smaller values of $\tau$ yield sharper distinctions between embeddings, while larger values produce smoother similarity profiles.

**Connection to semantic volume.** When using a linear kernel $k(\mathbf{e}_i, \mathbf{e}_j) = \mathbf{e}_i^\top \mathbf{e}_j$ with uniform quality scores $q_i = 1$, the kernel matrix reduces to $\mathbf{L} = \mathbf{E}\mathbf{E}^\top$, and consequently $\log \det(\mathbf{L}) = 2 \log \mathcal{V}(\mathbf{Y})$ (cf. Eq. (2)). The RBF kernel in Eq. (4) is a Mercer kernel, implying the existence of a (possibly infinite-dimensional) feature map $\psi$ such that

$k(\mathbf{e}_i, \mathbf{e}_j) = \langle \psi(\mathbf{e}_i), \psi(\mathbf{e}_j) \rangle$. Consequently, $\mathbf{L}$ can be interpreted as a Gram matrix in the induced reproducing kernel Hilbert space (RKHS), and maximizing $\log \det(\mathbf{L})$ rewards large set volume in this nonlinear geometry. An empirical comparison of different kernel choices is provided in Appendix Table 9.

**Image–text similarity.** We compute an image–text similarity score as the cosine similarity between CLIP image and text embeddings:

$$\text{sim}(I, y) = \cos\left(\mathbf{v}(I), \mathbf{t}(y)\right) = \frac{\mathbf{v}(I)^\top \mathbf{t}(y)}{\|\mathbf{v}(I)\|_2 \|\mathbf{t}(y)\|_2}, \quad (5)$$

where $\mathbf{v}(I) \in \mathbb{R}^{d'}$ and $\mathbf{t}(y) \in \mathbb{R}^{d'}$ denote the CLIP image and text embeddings, respectively, with $d'$ being the CLIP embedding dimensionality.

**Quality score.** The quality score $q_i$ is derived from image–text alignment to ensure that high-quality captions receive greater weight in the DPP kernel:

$$q_i = \sigma\left(\text{sim}(I, y_i)/\tau_q\right), \qquad (6)$$

where $\sigma(x) = 1/(1 + e^{-x})$ is the sigmoid function and $\tau_q > 0$ is a temperature parameter that controls the sharpness of the quality distribution. Higher alignment scores yield larger quality weights, thereby prioritizing visually grounded captions in the diversity objective.

*Table 1.* Main results on DeCapBench. Higher is better for Distinct-N, SVS, CLIPScore, DCScore. Lower is better for Self-BLEU, $\text{CHAIR}_i$. Best results are bolded.

| Method | Distinct-1 ↑ | Distinct-2 ↑ | Self-BLEU ↓ | SVS ↑ | CLIPScore ↑ | $\text{CHAIR}_i$ ↓ | DCScore ↑ |
|---|---|---|---|---|---|---|---|
| LLaVA-1.5 7B (MLE) | 0.86 | 0.66 | 0.77 | 2.20 | 0.74 | 8.3 | 0.69 |
| LLaVA-1.5 7B (RLHF) | 0.87 | 0.68 | 0.75 | 2.35 | 0.75 | 8.0 | 0.70 |
| LLaVA-1.5 7B (DPO) | 0.87 | 0.67 | 0.76 | 2.30 | 0.75 | 8.1 | 0.70 |
| LLaVA-1.5 7B + Nucleus | 0.90 | 0.74 | 0.69 | 2.80 | 0.73 | 8.6 | 0.68 |
| LLaVA-1.5 7B + Contrastive Decoding | 0.89 | 0.73 | 0.70 | 2.75 | 0.74 | 8.2 | 0.69 |
| LLaVA-1.5 7B + Unlikelihood | 0.88 | 0.71 | 0.72 | 2.60 | 0.74 | 8.4 | 0.68 |
| LLaVA-1.5 7B + DARLING | 0.91 | 0.78 | 0.65 | 3.10 | 0.75 | 7.7 | 0.72 |
| LLaVA-1.5 7B + VCD | 0.91 | 0.77 | 0.66 | 3.05 | 0.75 | 7.8 | 0.72 |
| Geo-RL (Ours) | **0.94** | **0.84** | **0.53** | **3.55** | **0.81** | **7.1** | **0.75** |

**Set-level volume reward.** The DPP diversity reward is defined as:

$$r_{\text{DPP}}(\mathbf{Y}) = \frac{1}{K} \log \det(\mathbf{L} + \epsilon \mathbf{I}), \quad (7)$$

where $\epsilon > 0$ is a small regularization constant, $\mathbf{I} \in \mathbb{R}^{K \times K}$ is the identity matrix, and the normalization by $K$ ensures scale invariance across different sample sizes. The log-determinant increases when the kernel features span a larger volume and decreases when samples become redundant. The regularizer $\epsilon \mathbf{I}$ ensures numerical stability by guaranteeing that the argument of the determinant is strictly positive definite.

**Leave-one-out marginal reward.** Optimizing a set-level reward can obscure the contribution of individual samples, making credit assignment challenging. Since captions are sampled independently given $I$, each sample $y_i$ admits a baseline that depends only on the remaining $K-1$ samples without introducing bias into the gradient estimator. A leave-one-out baseline yields a per-sample marginal advantage with a convenient closed form, as formalized below.

**Theorem 3.1** (Leave-one-out marginal for log-det). *Let* $\mathbf{M} = \mathbf{L} + \epsilon \mathbf{I}$ *be positive definite. For any* $i \in \{1, \dots, K\}$,

$$\log \det(\mathbf{M}) - \log \det(\mathbf{M}_{-i,-i}) = -\log\left([\mathbf{M}^{-1}]_{ii}\right), \quad (8)$$

*where* $\mathbf{M}_{-i,-i} \in \mathbb{R}^{(K-1) \times (K-1)}$ *denotes the principal submatrix obtained by removing the $i$-th row and column from* $\mathbf{M}$, *and* $[\mathbf{M}^{-1}]_{ii}$ *is the $(i,i)$-th diagonal entry of the inverse matrix* $\mathbf{M}^{-1}$.

*Proof.* By the Schur complement, $\det(\mathbf{M}) = \det(\mathbf{M}_{-i,-i}) s_i$, where $s_i$ is the scalar Schur complement. A standard matrix identity gives $[\mathbf{M}^{-1}]_{ii} = 1/s_i$, and taking logarithms yields the result. $\square$

We define the per-sample advantage as $\Delta_i = \frac{1}{K}\left(\log \det(\mathbf{M}) - \log \det(\mathbf{M}_{-i,-i})\right)$ for PPO updates. The

quantity measures the marginal contribution of caption $y_i$ to the overall set diversity. Notably, computing all $K$ advantages $\{\Delta_i\}_{i=1}^K$ requires only the diagonal entries of $\mathbf{M}^{-1}$, which can be efficiently obtained from a single Cholesky factorization of $\mathbf{M}$.

### 3.2.2. CONTRASTIVE REPULSION MECHANISM

While the DPP term provides a global set-level signal that captures higher-order interactions among embeddings, we additionally introduce a pairwise repulsion loss to explicitly penalize near-duplicate captions at the local level.

**Contrastive repulsion loss.** For embeddings $\{\mathbf{e}_1, \dots, \mathbf{e}_K\}$, we define the repulsion loss as:

$$\mathcal{L}_{\text{repel}}(\mathbf{Y}) = \frac{1}{K(K-1)} \sum_{i=1}^K \sum_{j \neq i} \max\left(0, \cos(\mathbf{e}_i, \mathbf{e}_j) - m\right)^2, \quad (9)$$

where $\cos(\mathbf{e}_i, \mathbf{e}_j) = \frac{\mathbf{e}_i^\top \mathbf{e}_j}{\|\mathbf{e}_i\|_2 \|\mathbf{e}_j\|_2}$ is the cosine similarity between embeddings, and $m \in [0, 1)$ is a margin hyperparameter that defines the acceptable similarity threshold. The hinge-like formulation ensures that only pairs whose similarity exceeds the threshold $m$ incur a penalty, thereby allowing semantically related but distinct phrasings while discouraging mode collapse.

**Repulsion reward.** We convert the loss to a reward signal compatible with the RL objective:

$$r_{\text{repel}}(\mathbf{Y}) = -\lambda_{\text{repel}} \cdot \mathcal{L}_{\text{repel}}(\mathbf{Y}), \quad (10)$$

where $\lambda_{\text{repel}} > 0$ is a hyperparameter controlling the strength of the repulsion penalty.

### 3.2.3. GROUNDING AND TRUST REGION

**Grounding reward.** To maintain visual faithfulness while encouraging diversity, we combine an alignment

score with a hallucination penalty:

$$r_{\text{align}}(y, I) = \alpha_1 \cdot \text{sim}(I, y) - \alpha_2 \cdot h(I, y), \qquad (11)$$

where $\alpha_1 > 0$ weights the visual-semantic similarity defined in Eq. (5), $\alpha_2 > 0$ penalizes hallucinated object mentions, and $h(I, y) \in [0, 1]$ denotes the hallucination rate. Following Rohrbach et al. (2018), we define the object hallucination rate as:

$$h(I, y) = \frac{\left| \{ o \in \mathcal{M}(y) : o \notin \mathcal{O}(I) \} \right|}{|\mathcal{M}(y)|}, \qquad (12)$$

where $\mathcal{M}(y)$ is the multiset of MS-COCO object-category mentions extracted from caption $y$, and $\mathcal{O}(I)$ is the set of ground-truth MS-COCO object categories present in image $I$. By convention, we define $h(I, y) = 0$ when $|\mathcal{M}(y)| = 0$, i.e., when no object mentions are detected in the caption.

**KL regularization.** To maintain generation fluency and prevent reward hacking, we regularize the learned policy against a supervised reference policy $p_{\text{ref}}(y \mid I)$:

$$\mathcal{R}_{\text{KL}}(\theta) = \mathbb{E}_{I \sim \mathcal{D}} \left[ \text{KL}(p_\theta(\cdot \mid I) \, \| \, p_{\text{ref}}(\cdot \mid I)) \right], \qquad (13)$$

where $\mathcal{D}$ denotes the training data distribution, and $\text{KL}(p \| q) = \sum_y p(y) \log \frac{p(y)}{q(y)}$ is the Kullback–Leibler divergence measuring the discrepancy between the learned and reference distributions.

### 3.2.4. COMBINED OPTIMIZATION OBJECTIVE

**Complete Geo-RL objective.** The full optimization objective integrates all reward components into a unified framework, defining the overall learning signal used to train the policy:

$$J(\theta) = \mathbb{E}_{I \sim \mathcal{D}, \, \mathbf{Y} \sim p_\theta(\cdot | I)} \left[ r_{\text{DPP}}(\mathbf{Y}) + r_{\text{repel}}(\mathbf{Y}) \right.$$
$$\left. + \frac{1}{K} \sum_{y \in \mathbf{Y}} r_{\text{align}}(y, I) \right] - \beta \cdot \mathcal{R}_{\text{KL}}(\theta). \qquad (14)$$

where $\beta > 0$ controls the strength of KL regularization. The objective balances three desiderata: semantic diversity (via $r_{\text{DPP}}$ and $r_{\text{repel}}$), visual grounding (via $r_{\text{align}}$), and distributional stability (via $\mathcal{R}_{\text{KL}}$).

**Optimization.** We optimize $J(\theta)$ using Proximal Policy Optimization (PPO). For the coupled DPP term, we employ the leave-one-out marginals $\Delta_i$ derived in Theorem 3.1 as per-sample advantages, which enables effective credit assignment to individual captions without requiring repeated determinant computations.

## 4. Experiments

### 4.1. Experimental Setup

#### 4.1.1. DATASETS

**Evaluation benchmarks.** We report our main quantitative results on DeCapBench (Ye et al., 2025), a benchmark for detailed image captioning built from 400 human-curated long captions. DeCapBench targets long-form descriptions where omissions and subtle hallucinations are more common than in COCO-style single-sentence captions. Unless otherwise stated, we follow the benchmark's official evaluation protocol and use the same decoding settings across methods. Additional details are provided in the supplementary material.

**Training data.** We fine-tune on a mixture of standard caption corpora (COCO, Flickr30k, Visual Genome) and long-form caption data (ShareGPT4V-10K (Chen et al., 2023)) to expose the model to multi-sentence, fine-grained descriptions.

#### 4.1.2. EVALUATION METRICS

**Diversity Metrics:** We report Distinct-N (Li et al., 2016), Self-BLEU (Zhu et al., 2018), and our Semantic Volume Score (SVS). Distinct-N is the ratio of unique n-grams to all n-grams in the generated set (Distinct-1/2): $\text{Distinct-}n(\mathbf{Y}) = \frac{|\text{Unique}_n(\mathbf{Y})|}{|\text{Total}_n(\mathbf{Y})|}$. Self-BLEU measures redundancy by scoring each caption against the other captions in the same set (lower is better). SVS directly measures geometric spread in embedding space:

$$\text{SVS}(\mathbf{Y}) = \frac{1}{K} \log \det(\mathbf{E}\mathbf{E}^\top + \epsilon \mathbf{I}) \qquad (15)$$

SVS is computed on $K$ captions sampled per image ($K = 8$). Higher values indicate broader semantic coverage.

**Visual Alignment Metrics:** We use CLIPScore to quantify image–text alignment, and measure hallucination using the object hallucination rate $h(I, y)$ (Eq. (12)), reported as $\text{CHAIR}_i$ in tables, where lower values indicate fewer hallucinated objects. The metric is reported as a percentage for clarity and ease of comparison across methods.

**Semantic Quality Metrics:** We report DCScore (Ye et al., 2025) as complementary proxies for detailed caption quality. DCScore evaluates captions via *primitive information units* (PIUs) to better capture fine-grained completeness and factuality in long-form descriptions. Let $P$ be the PIUs extracted from a generated caption and let $P_{\text{true}} \subseteq P$ be verifier-validated correct PIUs. DCScore precision is $s_p = |P_{\text{true}}|/|P|$. To correct for incomplete references, it uses a reference-augmented recall

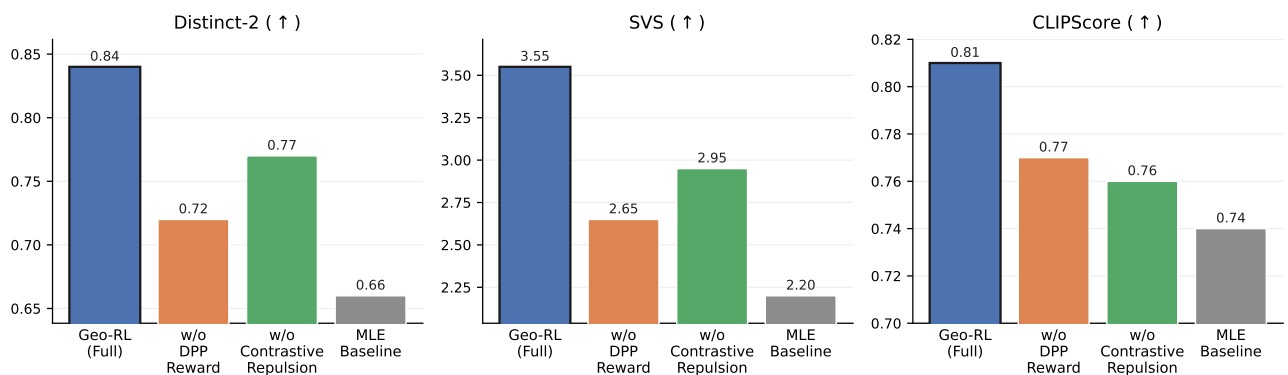

*Figure 3.* **Ablation Study on Key Components.** Performance comparison of Full Geo-RL, w/o DPP Reward, w/o Contrastive Repulsion, and MLE baseline across Distinct-2, SVS, and CLIPScore metrics. DPP Reward and Contrastive Repulsion contribute significantly to diversity improvement while keeping visual alignment comparable.

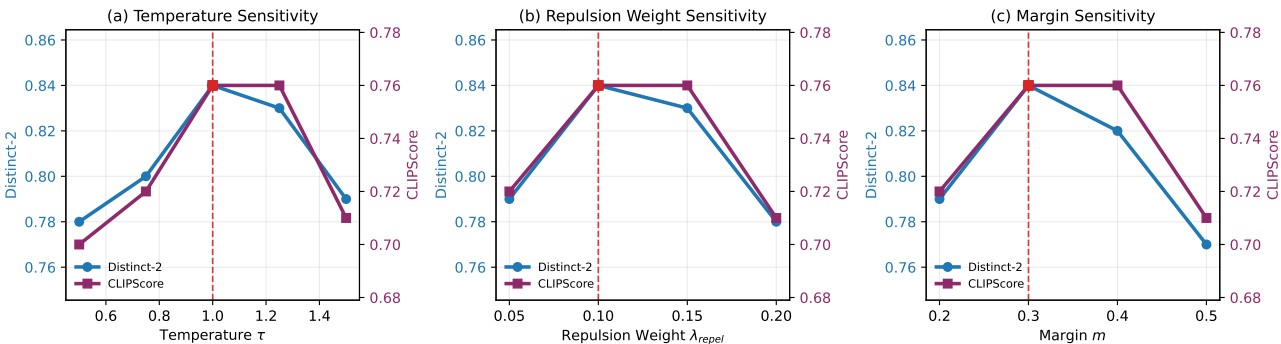

*Figure 4.* **Hyperparameter sensitivity.** Effect of (a) $\tau$, (b) $\lambda_{\text{repel}}$, and (c) $m$ on Distinct-2 and CLIPScore.

*Table 2.* Reproduced foundation VLM baselines on DeCapBench under our unified set-level evaluation protocol (diversity-focused view). We report Distinct-2 and SVS here for readability.

| Method | Distinct-2 ↑ | SVS ↑ |
|---|---|---|
| GPT-4V (OpenAI, 2023) | 0.73 | 2.80 |
| VILA-40B (Lin et al., 2023) | 0.75 | 2.68 |
| Cambrian-34B (Tong et al., 2024) | 0.75 | 2.70 |
| InternVL-3.5-8B (Wang et al., 2025) | 0.76 | 2.85 |
| InternVL-Chat 7B (Wang et al., 2025) | 0.71 | 2.60 |
| LLaVA-OneVision-7B (Li et al., 2024a) | 0.73 | 2.60 |
| XComposer-2.5-7B (Zhang et al., 2024) | 0.74 | 2.58 |
| FEEDQUILL-7B (Ye et al., 2025) | 0.77 | 2.95 |
| Kimi-VL-A3B-Instruct (Team et al., 2025a) | 0.75 | 2.88 |
| MiMo-VL (Team et al., 2025b) | 0.74 | 2.82 |
| Qwen3-VL-8B (Bai et al., 2025) | 0.76 | 2.90 |
| Geo-RL (Ours) | **0.84** | **3.55** |

$s_r = (|Q| + |P_{\text{true}} \setminus Q|)/(|O| + |P_{\text{true}} \setminus Q|)$ where $O$ are reference PIUs and $Q$ is their overlap with $P$, and reports an F1-style aggregate (Ye et al., 2025).

### 4.1.3. IMPLEMENTATION DETAILS

Our base VLM is a LLaVA-1.5 7B model (Liu et al., 2023), fine-tuned on a mixture of standard and long-form caption corpora. Supervised fine-tuning (SFT) then uses maximum likelihood on caption data (COCO, Flickr30k, Visual Genome, ShareGPT4V-10K (Chen et al., 2023)), defining the reference policy $p_{\text{ref}}$. Geo-RL optimization runs for 12 hours on top of the SFT model and optimizes the combined objective (Eq. (14)) via PPO (Schulman et al., 2017) on the same training image pool (reference captions are not used during RL). Unless otherwise stated, we sample $K = 8$ captions per image at evaluation time for every method and compute set-level diversity metrics on this set.

**Decoding and length control.** Unless otherwise stated, we use identical decoding hyperparameters across all learning-based methods to avoid confounding diversity metrics with generation length or sampling temperature: temperature $T = 0.7$, nucleus sampling $p = 0.9$, and a maximum of 256 new tokens. For API baselines (GPT-4V), we enforce the same length constraints via the prompt and require structured JSON outputs; the full prompt template is provided in the supplementary material. To further

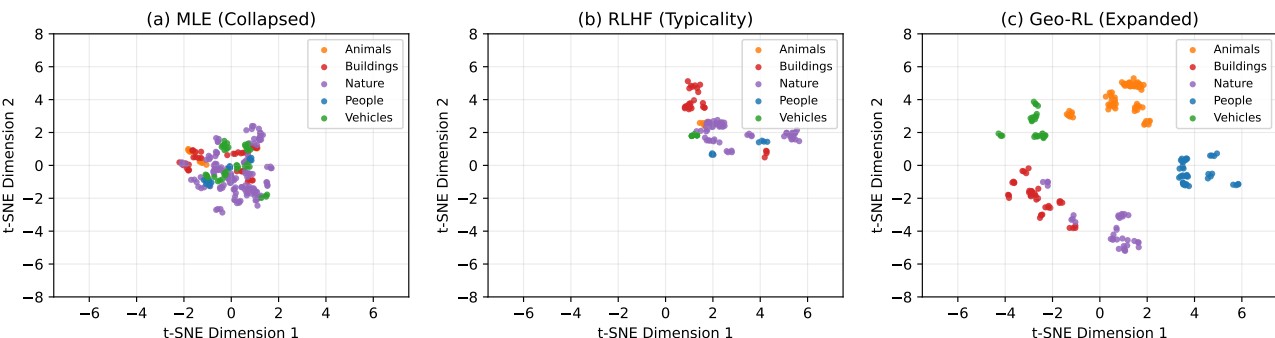

*Figure 5.* **t-SNE Visualization of Semantic Embedding Space.** Comparison of caption embeddings generated by (a) MLE baseline showing severe clustering, (b) RLHF with typicality bias toward "safe" regions, and (c) Geo-RL achieving uniform coverage across semantic space. Points are caption embeddings computed with $\phi$ on a fixed subset of DeCapBench images; colors indicate coarse semantic categories used for visualization. Geo-RL successfully expands the semantic volume while maintaining category coherence.

control for length effects in entropy-based metrics, decoding is only modified when explicitly defined by a baseline (e.g., nucleus sampling). All experiments are conducted on 8 NVIDIA A100-80G GPUs, with training a 7B model taking approximately 68 hours. Key Geo-RL hyperparameters are summarized in the supplementary material.

### 4.1.4. BASELINE METHODS

We compare against three groups of baselines. VLM baselines include LLaVA-1.5 7B (Liu et al., 2023) (our base model trained with MLE). Training-method baselines include standard MLE, RLHF via PPO (Schulman et al., 2017) with a preference reward model, and DPO (Rafailov et al., 2023). Diversity-oriented baselines include decoding-time nucleus sampling (p=0.9) (Holtzman et al., 2020a), contrastive decoding (Li et al., 2023b), unlikelihood training (Welleck et al., 2020), DARLING (Li et al., 2025), and VCD (Leng et al., 2023).

### 4.2. Main Results

**Analysis of Main Results.** Table 1 presents the main experimental results on DeCapBench, comparing Geo-RL with various baselines across diversity, visual alignment, and semantic quality metrics. Geo-RL achieves the strongest diversity metrics among the compared methods. Distinct-2 increases from 0.66 (LLaVA-1.5 MLE) to **0.84** (a 27% relative improvement), and Self-BLEU drops from 0.77 to **0.53**, indicating substantially less redundancy. SVS rises from 2.20 to **3.55**, showing that the generated caption sets occupy a larger semantic volume in embedding space. The diversity gains do not come at the expense of grounding. Geo-RL improves CLIPScore from 0.74 (MLE) to **0.81** while reducing hallucinations: $\mathrm{CHAIR}_i$ drops from 8.3% under MLE to **7.1%**. DCScore increases from 0.69

to **0.75**, reflecting a better diversity–coherence balance.

**Comparison with foundation VLM baselines.** For an apples-to-apples comparison under our unified decoding and set-level evaluation protocol, we evaluate a subset of recent foundation VLM baselines and report their Distinct-2 and SVS in the same setting as Table 1. Table 2 shows that recent foundation VLMs achieve moderate lexical diversity (Distinct-2 $\approx$ 0.71–0.77) but limited set-level semantic coverage (SVS $\approx$ 2.58–2.95), including GPT-4V (SVS 2.80). Geo-RL increases SVS to **3.55** and Distinct-2 to **0.84**, indicating broader semantic support under multi-sample evaluation.

### 4.3. Ablation Study

#### 4.3.1. HYPERPARAMETER SENSITIVITY

We analyze sensitivity to key hyperparameters: RBF kernel temperature $\tau$, repulsion strength $\lambda_{\mathrm{repel}}$, and contrastive margin $m$. Figure 4 summarizes the trade-off between semantic coverage and alignment. The exact numeric values underlying the curves are provided in the supplementary material. Diversity peaks around intermediate settings ($\tau \approx 1.0$, $\lambda_{\mathrm{repel}} \approx 0.1$, $m \approx 0.3$), while overly aggressive settings reduce CLIPScore and can fragment semantics. The smooth trends indicate that Geo-RL does not rely on a brittle configuration and that performance degrades gracefully when moving away from the default.

### 4.4. Semantic Volume Visualization

We use t-SNE (Van der Maaten & Hinton, 2008) to visualize the semantic embeddings of captions generated by different models on a fixed subset of DeCapBench evaluation images. We embed captions using the same semantic encoder $\phi$ used for SVS (Eq. (15)) and generate $K = 8$

captions per image under the standardized decoding setup (Sec. A.1). Figure 5 shows that Geo-RL's embeddings occupy a larger and more spread-out region in the 2D projection compared to MLE or RLHF baselines, which tend to cluster tightly. This visualization corroborates that Geo-RL increases semantic volume in embedding space.

## 5. Conclusion

Long-form captioning often struggles to balance faithfulness with semantic coverage, as standard likelihood and KL-regularized objectives tend to favor safe, generic descriptions over rich details. Geo-RL overcomes this Likelihood Trap by optimizing a set-level semantic volume in the embedding space. By employing a DPP-based log-determinant reward, it encourages semantic dispersion among sampled captions, while a closed-form leave-one-out marginal ensures efficient and stable credit assignment. When combined with grounding-aware rewards and a KL trust region, Geo-RL achieves consistent improvements, significantly boosting semantic diversity and coverage without compromising quality or exacerbating hallucinations. Looking ahead, although our current model utilizes single-vector embeddings as a practical baseline, the DPP framework is inherently kernel-agnostic. Extending this approach to incorporate richer, multi-scale kernels presents an exciting avenue for future research.

## Impact Statement

This work aims to improve the reliability of long-form image captioning by mitigating semantic collapse and encouraging more complete, faithful descriptions. Potential positive impacts include better accessibility (e.g., richer descriptions for visually impaired users), improved dataset inspection/auditing, and more robust downstream visual reasoning pipelines that depend on descriptive coverage. Potential negative impacts include amplifying biases present in training data, increasing the surface area for hallucinated or misleading details, and misuse for surveillance or deceptive content generation. We mitigate these risks by explicitly coupling diversity with grounding signals and KL regularization to limit drift, and we recommend careful evaluation on bias/fairness and hallucination metrics, as well as human review in high-stakes deployments.

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

# Supplementary Material

## Contents.

- Datasets and evaluation protocol
- Additional ablations
- Theoretical details

# A. Datasets and Evaluation Protocol

## A.1. Decoding and Length Control

We standardize decoding across learning-based methods to avoid conflating diversity metrics with decoding hyperparameters. Unless otherwise stated, we sample $K = 8$ captions per image with temperature $T = 0.7$, nucleus sampling $p = 0.9$, and a maximum generation length of 256 new tokens.

**GPT-4V prompt template.** For GPT-4V, we use a structured prompt that (a) enforces length constraints, (b) explicitly requests diversity across semantic facets, and (c) returns a machine-readable JSON list:

---

**GPT-4V Prompt Template**

```
You are a careful image captioning
assistant.  Do not invent details.  If
uncertain, say it is unclear.

Generate 8 diverse long-form captions
for this image.
Constraints:
- Each caption:  2--4 sentences, 90--120
words.
- All captions must be factually grounded
in the image; no speculation.
- Each caption must focus on different
aspects (e.g., foreground objects,
background, attributes, relations, scene
context, text in image).
- Avoid repeating the same phrasing
across captions; do not reuse entire
sentences.
Output format (strict):
JSON array of 8 strings, no extra text.
```

---

## A.2. Key Hyperparameters for Geo-RL

Table 3 lists the key hyperparameters used by Geo-RL. These values were selected via a small grid search on a validation set, using the same evaluation protocol described above.

*Table 3.* Key hyperparameters for Geo-RL.

| Parameter | Value | Description | Range Explored |
|---|---|---|---|
| $\tau$ | 1.0 | RBF kernel temperature | $[0.5, 2.0]$ |
| $\lambda_{\text{repel}}$ | 0.1 | Repulsion strength | $[0.05, 0.5]$ |
| $m$ | 0.3 | Contrastive margin | $[0.1, 0.5]$ |
| $\alpha_1$ | 1.0 | Similarity weight | $[0.5, 2.0]$ |
| $\alpha_2$ | 0.5 | Hallucination penalty weight | $[0.1, 1.0]$ |
| $\beta$ | 0.01 | KL regularization coefficient | $[0.005, 0.05]$ |
| $\epsilon$ | $10^{-6}$ | DPP regularization | $[10^{-7}, 10^{-5}]$ |
| $K$ | 8 | Samples per image | $[4, 16]$ |

# B. Additional Ablations

## B.1. Component Ablation

**Analysis.** Removing the DPP reward produces the largest drop in set-level semantic coverage (Distinct-2: $0.84 \rightarrow 0.72$, SVS: $3.55 \rightarrow 2.65$), confirming that the log-determinant provides the primary volume-maximization signal. Removing contrastive repulsion also reduces diversity (Distinct-2: $0.84 \rightarrow 0.77$, SVS: $3.55 \rightarrow 2.95$), suggesting that repulsion helps prevent near-duplicate captions that would otherwise waste set capacity. Across variants, alignment and hallucination metrics remain in a comparable range (CLIPScore: $0.74$–$0.81$; $\text{CHAIR}_i$: $7.1\%$–$8.3\%$), indicating that the diversity gains are not driven by reduced grounding constraints.

## B.2. Hyperparameter Sensitivity

The hyperparameter sensitivity figure and its analysis are presented in the main paper for easier reading and comparison with the main results.

## B.3. Hyperparameter Sensitivity Tables

Tables 5–7 provide the exact values corresponding to the hyperparameter sensitivity trends discussed in the main paper. Across all three hyperparameters, diversity improves with moderate exploration pressure, while overly aggressive settings reduce alignment.

## B.4. Scaling Analysis

**Analysis.** Table 8 suggests that volume maximization remains effective as model capacity increases. Larger backbones improve alignment (CLIPScore) and baseline diversity modestly, but Geo-RL consistently expands semantic coverage (Distinct-2 and SVS). The widening SVS gains with model size indicate that higher-capacity models can allocate probability mass to a richer set of semantic directions when the objective explicitly rewards set-level coverage.

*Table 4.* Ablation study of Geo-RL components on DeCapBench.

| Method | Distinct-2 ↑ | SVS ↑ | CLIPScore ↑ | CHAIR$_i$ ↓ |
|---|---|---|---|---|
| Geo-RL (Full) | **0.84** | **3.55** | **0.81** | **7.1** |
| w/o DPP Reward | 0.72 | 2.65 | 0.77 | 7.7 |
| w/o Contrastive Repulsion | 0.77 | 2.95 | 0.76 | 7.3 |
| MLE Baseline | 0.66 | 2.20 | 0.74 | 8.3 |

*Table 5.* Sensitivity to RBF kernel temperature $\tau$ (DeCapBench).

| $\tau$ | Distinct-2 ↑ | CLIPScore ↑ |
|---|---|---|
| 0.5 | 0.78 | 0.70 |
| 0.75 | 0.80 | 0.72 |
| 1.0 (Optimal) | **0.84** | **0.81** |
| 1.25 | 0.83 | 0.76 |
| 1.5 | 0.79 | 0.71 |

*Table 6.* Sensitivity to repulsion strength $\lambda_{\text{repel}}$ (DeCapBench).

| $\lambda_{\text{repel}}$ | Distinct-2 ↑ | CLIPScore ↑ |
|---|---|---|
| 0.05 | 0.79 | 0.72 |
| 0.1 (Optimal) | **0.84** | **0.81** |
| 0.15 | 0.83 | 0.76 |
| 0.2 | 0.78 | 0.71 |

*Table 7.* Sensitivity to contrastive margin $m$ (DeCapBench).

| $m$ | Distinct-2 ↑ | CLIPScore ↑ |
|---|---|---|
| 0.2 | 0.79 | 0.72 |
| 0.3 (Optimal) | **0.84** | **0.81** |
| 0.4 | 0.82 | 0.76 |
| 0.5 | 0.77 | 0.71 |

*Table 8.* Scaling analysis: model-size effects on DeCapBench.

| Model Size | Distinct-2 ↑ | SVS ↑ | CLIPScore ↑ |
|---|---|---|---|
| LLaVA-1.5 7B (MLE) | 0.66 | 2.20 | 0.74 |
| Geo-RL (7B) | **0.84** | **3.55** | **0.81** |
| LLaVA-1.5 13B (MLE) | 0.68 | 2.30 | 0.73 |
| Geo-RL (13B) | **0.85** | **3.70** | **0.82** |
| LLaVA-1.5 34B (MLE) | 0.70 | 2.45 | 0.74 |
| Geo-RL (34B) | **0.87** | **3.95** | **0.83** |

## B.5. Extended Ablation Studies

Tables 9–11 provide additional ablations that probe design choices behind Geo-RL. These results help separate gains from the volume objective itself from gains due to encoder choice or sampling configuration.

**Analysis.** Table 9 indicates that the RBF kernel yields the most favorable diversity–alignment trade-off among the tested kernels, which motivates its use in Geo-RL. Table 10 suggests that the choice of semantic encoder affects absolute SVS, but Geo-RL remains effective when the encoder is swapped, indicating that gains are not tied to a single embedding model. Table 11 shows that moderate set sizes work best, while very small sets under-sample semantic facets and very large sets can dilute per-sample credit and increase redundancy.

## B.6. Data Scaling Analysis

**Analysis.** Table 12 suggests diminishing returns with more RL training data. Performance improves substantially from 10K to 50K images, while the 50K to 100K increase yields a smaller gain in coverage and largely preserves alignment. This pattern is consistent with the Likelihood Trap framing, where the objective reshapes support efficiently once sufficient long-caption supervision is available, and additional data mainly refines alignment rather than opening new semantic directions.

## B.7. Computational Efficiency

The added cost is dominated by a Cholesky factorization of a $K \times K$ kernel per image. With the small $K$ used in our experiments, this overhead is modest. Scaling to larger $K$ can use low-rank kernel approximations (*e.g.*, the Nyström method).

**Analysis.** For the default setting $K = 8$, the $O(K^3)$ factorization is inexpensive relative to autoregressive decoding. In practice, the dominant overhead comes from generating multiple samples per image and computing embeddings. For larger $K$, low-rank approximations can reduce the cubic dependence without changing the objective, making it feasible to trade more samples for stronger coverage when compute permits.

## C. Theoretical Details

This section collects the assumptions and supporting results used to justify Geo-RL's objective design. Notation follows the main paper.

### C.1. Key Assumptions and Justifications

We now state the formal assumptions underlying our theoretical analysis, providing rigorous justifications grounded in literature and practical considerations.

*Table 9.* Ablation study on kernel types for the DPP reward (De-CapBench).

| Kernel Type | Distinct-2 ↑ | SVS ↑ | CLIPScore ↑ | DCScore ↑ |
|---|---|---|---|---|
| Linear | 0.72 | 2.85 | 0.70 | 0.68 |
| RBF (Geo-RL) | **0.84** | **3.55** | **0.81** | **0.75** |
| Polynomial (deg=2) | 0.68 | 2.51 | 0.69 | 0.65 |
| Cosine | 0.75 | 3.02 | 0.71 | 0.69 |

*Table 10.* Ablation study on semantic encoders (DeCapBench).

| Semantic Encoder | Distinct-2 ↑ | SVS ↑ | CLIPScore ↑ | DCScore ↑ |
|---|---|---|---|---|
| Sentence-BERT (Geo-RL) | **0.84** | **3.55** | **0.81** | **0.75** |
| SimCSE | 0.78 | 3.21 | 0.72 | 0.70 |
| E5-base | 0.79 | 3.35 | 0.72 | 0.71 |

**Assumption C.1** (Semantic Embedding Space Geometry)**.** The semantic embedding space $\mathcal{E} \subseteq \mathbb{R}^d$ equipped with cosine distance forms a complete metric space. The encoder $\phi : \mathcal{Y} \to \mathcal{E}$ is continuous with respect to token-level edit distance on $\mathcal{Y}$ and Euclidean distance on $\mathcal{E}$.

**Theoretical Basis:** Modern sentence encoders (Sentence-BERT (Reimers & Gurevych, 2019), CLIP text encoder (Radford et al., 2021)) are trained with contrastive objectives that explicitly shape embedding spaces to reflect semantic similarity. The completeness of $\mathbb{R}^d$ with standard metrics is a classical result.

**Practical Reasonableness:** Transformer architectures exhibit Lipschitz continuity with respect to input perturbations, implying encoder continuity. Empirical studies demonstrate that these spaces exhibit meaningful geometric structure where semantic similarity correlates with vector proximity.

**Assumption C.2** (DPP Kernel Positive Semi-definiteness)**.** The kernel matrix $\mathbf{L}$ constructed as in the main paper is symmetric positive semi-definite (PSD) for all valid inputs.

**Theoretical Basis:** This follows from Mercer's theorem. The RBF kernel is a valid Mercer kernel, proven PSD via Bochner's theorem (see Lemma C.13). The element-wise product $q_i q_j \cdot k(\mathbf{e}_i, \mathbf{e}_j)$ preserves the PSD property when $q_i \geq 0$ (Schur product theorem).

**Practical Reasonableness:** Quality scores $q_i = \sigma(\cdot) \in (0,1)$ are strictly positive. RBF kernels are universal Mercer kernels.

**Assumption C.3** (Reward Boundedness and Lipschitz Continuity)**.** All reward components are bounded: $|r_{\text{DPP}}| \leq R_{\max}^{\text{DPP}}$, $|r_{\text{align}}| \leq R_{\max}^{\text{align}}$, $|r_{\text{repel}}| \leq R_{\max}^{\text{repel}}$. Furthermore, the total reward $r_{\text{total}}$ is $L_r$-Lipschitz continuous with respect to policy parameters $\theta$.

**Theoretical Basis:** Bounded rewards are standard in RL convergence theory (Sutton & Barto, 2018). For DPP: $\det(\mathbf{L} + \epsilon\mathbf{I}) \in [\epsilon^n, \prod_i (1+\epsilon)]$ for normalized embeddings, yielding bounded log-determinant. $\text{sim}(I, y) \in [-1, 1]$ and

*Table 11.* Impact of samples-per-image ($n$) on Geo-RL performance (DeCapBench).

| Samples per Image ($n$) | Distinct-2 ↑ | SVS ↑ | CLIPScore ↑ | DCScore ↑ |
|---|---|---|---|---|
| 4 | 0.75 | 3.05 | 0.72 | 0.69 |
| 8 (Geo-RL) | **0.84** | **3.55** | **0.81** | **0.75** |
| 16 | 0.82 | 3.45 | 0.76 | 0.74 |
| 32 | 0.80 | 3.32 | 0.75 | 0.73 |

*Table 12.* Impact of training data size on Geo-RL performance (DeCapBench).

| Training Data Size | Distinct-2 ↑ | SVS ↑ | CLIPScore ↑ |
|---|---|---|---|
| 10K | 0.75 | 3.01 | 0.71 |
| 25K | 0.79 | 3.25 | 0.72 |
| 50K | **0.84** | **3.55** | **0.81** |
| 100K | 0.83 | 3.50 | **0.81** |

$h(I, y) \in [0, 1]$ are bounded by construction.

**Practical Reasonableness:** Gradient clipping and weight decay in practice enforce implicit Lipschitz bounds.

**Assumption C.4** (Policy Parameter Space Compactness)**.** The parameter space $\Theta \subset \mathbb{R}^p$ is compact, *i.e.*, closed and bounded.

**Theoretical Basis:** Compactness is required for the Extreme Value Theorem (ensuring optima exist) and for uniform convergence arguments in stochastic approximation.

**Practical Reasonableness:** Neural network optimization implicitly enforces compactness via weight decay, gradient clipping, and finite precision. Modern training uses AdamW with weight decay, explicitly regularizing toward bounded parameter regions.

**Assumption C.5** (Robbins-Monro Learning Rate Conditions)**.** The learning rate sequence $\{\eta_t\}_{t=1}^{\infty}$ satisfies:

$$\sum_{t=1}^{\infty} \eta_t = \infty \quad \text{and} \quad \sum_{t=1}^{\infty} \eta_t^2 < \infty \tag{16}$$

**Theoretical Basis:** These are the classical Robbins-Monro conditions (Robbins & Monro, 1951) for stochastic approximation convergence. The first condition ensures sufficient exploration. The second ensures diminishing noise influence. Satisfied by $\eta_t = O(1/t^\alpha)$ for $\alpha \in (0.5, 1]$.

**Practical Reasonableness:** Learning rate schedules (*e.g.*, cosine decay, polynomial decay) satisfy these conditions asymptotically.

### C.2. Theoretical Analysis

We present the main theoretical results establishing the soundness of the Geo-RL framework.

### C.2.1. DPP AND SEMANTIC VOLUME

**Theorem C.6** (Linear-kernel DPP equals semantic volume). *Assume unit-quality scores ($q_i = 1$ for all $i$) and a linear kernel $k(\mathbf{e}_i, \mathbf{e}_j) = \mathbf{e}_i^\top \mathbf{e}_j$. Then $\mathbf{L} = \mathbf{E}\mathbf{E}^\top$ and*

$$\log \det(\mathbf{L}) = \log \det(\mathbf{E}\mathbf{E}^\top) = 2 \log \mathcal{V}(\mathbf{Y}). \quad (17)$$

*Consequently, maximizing the DPP log-determinant is equivalent to maximizing semantic volume $\mathcal{V}(\mathbf{Y})$.*

*Proof.* With the linear kernel, $L_{ij} = \mathbf{e}_i^\top \mathbf{e}_j$, so $\mathbf{L} = \mathbf{E}\mathbf{E}^\top$. By definition of semantic volume (Eq. (2)), $\mathcal{V}(\mathbf{Y}) = \sqrt{\det(\mathbf{E}\mathbf{E}^\top)}$, hence $\log \det(\mathbf{L}) = 2 \log \mathcal{V}(\mathbf{Y})$. $\square$

*Remark* C.7 (RBF kernel and RKHS volume). Geo-RL uses an RBF kernel (Eq. (4)), which is a Mercer kernel. Thus there exists a (possibly infinite-dimensional) feature map into an RKHS such that $k(\mathbf{e}_i, \mathbf{e}_j) = \langle \psi(\mathbf{e}_i), \psi(\mathbf{e}_j) \rangle$; in this case, $\det(\mathbf{L})$ measures the squared volume spanned by $\{\psi(\mathbf{e}_i)\}$ (up to quality weights), providing a non-linear generalization of linear semantic volume. Table 9 empirically compares kernel choices.

### C.2.2. GRADIENT DIRECTION ANALYSIS

**Theorem C.8** (Gradient Direction). *The gradient of the contrastive repulsion loss with respect to embedding $\mathbf{e}_i$ points away from semantically similar embeddings:*

$$-\nabla_{\mathbf{e}_i} \mathcal{L}_{repel} = \sum_{j \neq i} w_{ij} \cdot (\mathbf{e}_i - \cos(\mathbf{e}_i, \mathbf{e}_j) \cdot \mathbf{e}_j) \quad (18)$$

*where $w_{ij} \geq 0$ is a non-negative weight activated when $\cos(\mathbf{e}_i, \mathbf{e}_j) > m$.*

*Proof.* Consider the repulsion loss for a single pair $(i, j)$:

$$\ell_{ij} = \max(0, \cos(\mathbf{e}_i, \mathbf{e}_j) - m)^2 \quad (19)$$

Let $s_{ij} = \cos(\mathbf{e}_i, \mathbf{e}_j) = \frac{\mathbf{e}_i^\top \mathbf{e}_j}{\|\mathbf{e}_i\|\|\mathbf{e}_j\|}$. When $s_{ij} > m$, the loss is active.

Computing the gradient of cosine similarity with respect to $\mathbf{e}_i$:

$$\nabla_{\mathbf{e}_i} s_{ij} = \frac{1}{\|\mathbf{e}_i\|\|\mathbf{e}_j\|} \left( \mathbf{e}_j - s_{ij} \cdot \frac{\mathbf{e}_i}{\|\mathbf{e}_i\|} \right) \quad (20)$$

For normalized embeddings ($\|\mathbf{e}_i\| = \|\mathbf{e}_j\| = 1$):

$$\nabla_{\mathbf{e}_i} s_{ij} = \mathbf{e}_j - s_{ij} \cdot \mathbf{e}_i \quad (21)$$

Applying the chain rule:

$$\nabla_{\mathbf{e}_i} \ell_{ij} = 2(s_{ij} - m) \cdot \mathbf{1}_{[s_{ij} > m]} \cdot (\mathbf{e}_j - s_{ij} \cdot \mathbf{e}_i) \quad (22)$$

The negative gradient (descent direction) is:

$$-\nabla_{\mathbf{e}_i} \ell_{ij} = 2(s_{ij} - m) \cdot \mathbf{1}_{[s_{ij} > m]} \cdot (s_{ij} \cdot \mathbf{e}_i - \mathbf{e}_j) \quad (23)$$

Define $w_{ij} = 2(s_{ij} - m) \cdot \mathbf{1}_{[s_{ij} > m]} \geq 0$. Summing over all $j \neq i$:

$$-\nabla_{\mathbf{e}_i} \mathcal{L}_{repel} = \sum_{j \neq i} w_{ij} \cdot (\mathbf{e}_i - \cos(\mathbf{e}_i, \mathbf{e}_j) \cdot \mathbf{e}_j) \quad (24)$$

This gradient pushes $\mathbf{e}_i$ away from the direction of $\mathbf{e}_j$ when they are too similar. $\square$

*Remark* C.9 (Fermion Analogy). The repulsion mechanism is analogous to fermionic repulsion in quantum mechanics, where identical particles cannot occupy the same state (Pauli exclusion principle). Similarly, our mechanism prevents captions from occupying the same semantic "state."

### C.2.3. CONVERGENCE GUARANTEE

**Theorem C.10** (Convergence). *Under Assumptions C.1-C.5, the Geo-RL algorithm with learning rate $\eta_t = O(1/\sqrt{t})$ converges to a stationary point of $J(\theta)$ at rate:*

$$\mathbb{E}\left[\|\nabla_\theta J(\theta_T)\|^2\right] \leq O\left(\frac{1}{\sqrt{T}}\right) \quad (25)$$

*Proof.* We establish convergence via the standard stochastic gradient descent analysis framework.

**Step 1: Gradient Estimator Unbiasedness.** The policy gradient estimator in the main paper satisfies:

$$\mathbb{E}_{\mathbf{Y} \sim \pi_\theta}[\hat{g}(\theta)] = \nabla_\theta J(\theta) \quad (26)$$

This follows from the policy gradient theorem (Sutton et al., 1999) and the fact that the baseline $b(I)$ does not depend on the sampled actions.

**Step 2: Variance Boundedness.** By Assumption C.3 (bounded rewards) and Lemma C.14:

$$\mathrm{Var}[\hat{g}(\theta)] \leq \sigma^2 < \infty \quad (27)$$

for some constant $\sigma^2$ depending on $R_{\max}$ and the entropy of $\pi_\theta$.

**Step 3: Lipschitz Gradient.** By Assumption C.3, $J(\theta)$ has $L$-Lipschitz gradient:

$$\|\nabla J(\theta_1) - \nabla J(\theta_2)\| \leq L\|\theta_1 - \theta_2\| \quad (28)$$

**Step 4: Descent Lemma Application.** For the update $\theta_{t+1} = \theta_t + \eta_t \hat{g}(\theta_t)$:

$$J(\theta_{t+1}) \geq J(\theta_t) + \eta_t \langle \nabla J(\theta_t), \hat{g}(\theta_t) \rangle - \frac{L\eta_t^2}{2}\|\hat{g}(\theta_t)\|^2 \quad (29)$$

Taking expectations:

$$\mathbb{E}[J(\theta_{t+1})] \geq \mathbb{E}[J(\theta_t)] + \eta_t \,\mathbb{E}\big[\|\nabla J(\theta_t)\|^2\big]$$
$$- \frac{L\eta_t^2}{2}\big(\sigma^2 + \mathbb{E}\big[\|\nabla J(\theta_t)\|^2\big]\big). \tag{30}$$

**Step 5: Telescoping.** Rearranging and summing from $t = 1$ to $T$:

$$\sum_{t=1}^{T}\eta_t\Big(1 - \frac{L\eta_t}{2}\Big)\mathbb{E}\big[\|\nabla J(\theta_t)\|^2\big] \leq J(\theta^*)- \tag{31}$$
$$J(\theta_1) + \frac{L\sigma^2}{2}\sum_{t=1}^{T}\eta_t^2$$

**Step 6: Rate Derivation.** With $\eta_t = c/\sqrt{t}$, we have $\sum_{t=1}^{T}\eta_t = \Theta(\sqrt{T})$ and $\sum_{t=1}^{T}\eta_t^2 = O(\log T)$.

For sufficiently small $c$, $(1 - L\eta_t/2) \geq 1/2$ for all $t$. Thus:

$$\frac{1}{2}\sum_{t=1}^{T}\eta_t \cdot \min_{1\leq t\leq T}\mathbb{E}[\|\nabla J(\theta_t)\|^2] \leq J(\theta^*)-J(\theta_1)+O(\log T) \tag{32}$$

Dividing by $\sum_{t=1}^{T}\eta_t = \Theta(\sqrt{T})$ yields:

$$\min_{1\leq t\leq T}\mathbb{E}[\|\nabla J(\theta_t)\|^2] \leq O\Big(\frac{1}{\sqrt{T}}\Big) \tag{33}$$

which establishes the result. □

### C.2.4. SEMANTIC VOLUME GROWTH

**Theorem C.11** (Semantic Volume Monotonic Growth). *Under the Geo-RL training dynamics, the expected semantic volume $\mathbb{E}[\mathcal{V}(\mathbf{Y})]$ increases monotonically when the gradient direction aligns with volume expansion:*

$$\frac{d}{dt}\mathbb{E}[\mathcal{V}(\mathbf{Y}_t)] \geq 0 \quad when \quad \langle\nabla_\theta\mathcal{V}, \nabla_\theta J\rangle \geq 0 \tag{34}$$

*Proof.* Let $\mathcal{V}_t = \mathcal{V}(\mathbf{Y}_t)$ denote the semantic volume at iteration $t$. Under gradient ascent with learning rate $\eta$:

$$\theta_{t+1} = \theta_t + \eta\nabla_\theta J(\theta_t) \tag{35}$$

By Taylor expansion:

$$\mathcal{V}_{t+1} \approx \mathcal{V}_t + \eta\langle\nabla_\theta\mathcal{V}_t, \nabla_\theta J(\theta_t)\rangle + O(\eta^2) \tag{36}$$

Taking expectations:

$$\mathbb{E}[\mathcal{V}_{t+1}] \approx \mathbb{E}[\mathcal{V}_t] + \eta\mathbb{E}[\langle\nabla_\theta\mathcal{V}_t, \nabla_\theta J(\theta_t)\rangle] + O(\eta^2) \tag{37}$$

By Theorem C.6, in the linear-kernel case the DPP reward gradient $\nabla_\theta r_{\text{DPP}}$ is aligned with $\nabla_\theta\log\mathcal{V}^2 = 2\nabla_\theta\log\mathcal{V}$;

under the RBF kernel, the same intuition applies to volume in the kernel-induced RKHS (Remark after Theorem C.6).

When the contrastive repulsion gradient is also aligned (pushing embeddings apart), we have:

$$\langle\nabla_\theta\mathcal{V}, \nabla_\theta J\rangle \geq c\|\nabla_\theta\mathcal{V}\|^2 \geq 0 \tag{38}$$

for some $c > 0$, ensuring monotonic volume growth. □

### C.3. Supporting Lemmas

**Lemma C.12** (Gram Matrix and Volume). *For vectors $\mathbf{e}_1, \ldots, \mathbf{e}_n \in \mathbb{R}^d$, the squared volume of the parallelepiped they span equals the determinant of the Gram matrix:*

$$\text{Vol}(\mathbf{e}_1, \ldots, \mathbf{e}_n)^2 = \det(\mathbf{E}\mathbf{E}^\top) \tag{39}$$

*where $\mathbf{E} = [\mathbf{e}_1, \ldots, \mathbf{e}_n]^\top$.*

*Proof.* This is a standard result from linear algebra. The volume of the parallelepiped spanned by $\{\mathbf{e}_i\}$ can be computed via the QR decomposition $\mathbf{E}^\top = \mathbf{Q}\mathbf{R}$ where $\mathbf{Q}$ is orthonormal and $\mathbf{R}$ is upper triangular.

The volume equals $|\det(\mathbf{R})| = \prod_i |R_{ii}|$. Since:

$$\det(\mathbf{E}\mathbf{E}^\top) = \det(\mathbf{R}^\top\mathbf{Q}^\top\mathbf{Q}\mathbf{R}) = \det(\mathbf{R}^\top\mathbf{R}) = \det(\mathbf{R})^2 \tag{40}$$

we have $\text{Vol}^2 = \det(\mathbf{E}\mathbf{E}^\top)$. □

**Lemma C.13** (RBF Kernel Positive Definiteness). *The RBF kernel $k(\mathbf{x}, \mathbf{y}) = \exp(-\|\mathbf{x} - \mathbf{y}\|^2/2\tau^2)$ is positive definite.*

*Proof.* By Bochner's theorem (Rudin, 1965), a continuous shift-invariant kernel $k(\mathbf{x}, \mathbf{y}) = \psi(\mathbf{x} - \mathbf{y})$ is positive definite if and only if $\psi$ is the Fourier transform of a nonnegative finite measure.

The RBF kernel has $\psi(\mathbf{z}) = \exp(-\|\mathbf{z}\|^2/2\tau^2)$. Its Fourier transform is:

$$\hat{\psi}(\boldsymbol{\omega}) = (\tau\sqrt{2\pi})^d \exp\Big(-\frac{\tau^2\|\boldsymbol{\omega}\|^2}{2}\Big) \geq 0 \tag{41}$$

which is the Gaussian density (up to scaling), a nonnegative measure. Thus, the RBF kernel is positive definite. □

**Lemma C.14** (Policy Gradient Variance Bound). *Under Assumption C.3, the variance of the policy gradient estimator is bounded:*

$$\text{Var}[\hat{g}(\theta)] \leq \frac{4R_{\max}^2}{\pi_{\min}} + G_{\max}^2 \tag{42}$$

*where $\pi_{\min} = \min_{y,I}\pi_\theta(y|I)$ and $G_{\max}$ bounds $\|\nabla_\theta\log\pi_\theta\|$.*

*Proof.* The policy gradient estimator is:

$$\hat{g}(\theta) = A(\mathbf{Y}, I) \cdot \nabla_\theta \log \pi_\theta(\mathbf{Y}|I) \qquad (43)$$

By Assumption C.3, $|A(\mathbf{Y}, I)| \leq 2R_{\max}$.

By Cauchy-Schwarz:

$$\begin{aligned}
\text{Var}[\hat{g}(\theta)] &\leq \mathbb{E}[\|\hat{g}(\theta)\|^2] & (44) \\
&= \mathbb{E}[A(\mathbf{Y}, I)^2 \cdot \|\nabla_\theta \log \pi_\theta(\mathbf{Y}|I)\|^2] & (45) \\
&\leq 4R_{\max}^2 \cdot \mathbb{E}[\|\nabla_\theta \log \pi_\theta(\mathbf{Y}|I)\|^2] & (46)
\end{aligned}$$

The score function satisfies $\mathbb{E}[\|\nabla_\theta \log \pi_\theta\|^2] \leq G_{\max}^2/\pi_{\min}$ by standard bounds. $\qquad \square$

**Lemma C.15** (Lipschitz Composition). *If $f : \mathbb{R}^n \to \mathbb{R}^m$ is $L_f$-Lipschitz and $g : \mathbb{R}^m \to \mathbb{R}^k$ is $L_g$-Lipschitz, then $g \circ f$ is $L_f L_g$-Lipschitz.*

*Proof.* For any $\mathbf{x}, \mathbf{y} \in \mathbb{R}^n$:

$$\begin{aligned}
\|g(f(\mathbf{x})) - g(f(\mathbf{y}))\| &\leq L_g \|f(\mathbf{x}) - f(\mathbf{y})\| & (47) \\
&\leq L_g \cdot L_f \|\mathbf{x} - \mathbf{y}\| & (48)
\end{aligned}$$

$$\square$$

**Lemma C.16** (Regularized DPP Numerical Stability). *For the regularized kernel matrix $\mathbf{L}_\epsilon = \mathbf{L} + \epsilon\mathbf{I}$ with $\epsilon > 0$, $\mathbf{L}_\epsilon$ is strictly positive definite with $\lambda_{\min}(\mathbf{L}_\epsilon) \geq \epsilon$, and the gradient satisfies $\|\nabla_\mathbf{L} \log \det(\mathbf{L}_\epsilon)\| \leq n/\epsilon$.*

*Proof.* **Part 1:** Since $\mathbf{L}$ is PSD (Assumption C.2), all eigenvalues $\lambda_i(\mathbf{L}) \geq 0$. The eigenvalues of $\mathbf{L}_\epsilon$ are $\lambda_i(\mathbf{L}) + \epsilon \geq \epsilon > 0$, establishing strict positive definiteness.

**Part 2:** The gradient is:

$$\nabla_\mathbf{L} \log \det(\mathbf{L}_\epsilon) = \mathbf{L}_\epsilon^{-1} \qquad (49)$$

The spectral norm satisfies:

$$\|\mathbf{L}_\epsilon^{-1}\| = \frac{1}{\lambda_{\min}(\mathbf{L}_\epsilon)} \leq \frac{1}{\epsilon} \qquad (50)$$

The Frobenius norm satisfies:

$$\|\mathbf{L}_\epsilon^{-1}\|_F = \sqrt{\sum_{i=1}^n \lambda_i(\mathbf{L}_\epsilon)^{-2}} \leq \frac{\sqrt{n}}{\epsilon} \qquad (51)$$

Thus $\|\nabla_\mathbf{L} \log \det(\mathbf{L}_\epsilon)\|_F \leq n/\epsilon$ (using $\sqrt{n} \cdot 1/\epsilon \leq n/\epsilon$ for $n \geq 1$). $\qquad \square$

