# OpenReview forum: "Escaping the Likelihood Trap: Geometric Diversity Optimization for Long-Form Image Captioning"
_ICML.cc/2026/Conference — ICML 2026 regular_

### Official Review · Reviewer_SEPx · 2026-02-27

**Soundness:** 3
**Presentation:** 3
**Significance:** 2
**Originality:** 2
**Overall Recommendation:** 4
**Confidence:** 4

**Summary:**

This paper aims at improving the diversity of VLM-based captioning models via reinforcement learning. Different from common methods like dense captions, which primarily focus on the semantics of captioning data, the authors incorporate both semantic correctness and diversity into the reward design and utilize the DDP kernel as the implementation. Extensive experiments demonstrate the effectiveness of the proposed method.

**Compliance With Llm Reviewing Policy:**

Affirmed.

**Final Justification:**

The rebuttal addresses most of my concerns. I choose to raise the score,

**Key Questions For Authors:**

As stated in the Weakness part, the controversial problem of this paper lies in the purpose of this paper: to develop a better captioner (c.f., paper tile) or a better VLM (c.f., paper abstract). If the latter, I think the authors need to further discuss the connection between diverse captions and better VLM capability.

**Limitations:**

Again, my main concern lies in the correlation between diverse captions and better VLMs.

**Strengths And Weaknesses:**

- Strengths
  - This paper focuses on caption diversity as another key factor besides semantic correctness.
  - The proposed method is quite straightforward.
  - This paper has a solid theoretical background.
- Weakness
  - Novelty will be a controversial problem in this paper.
    - Although a fancy story, the main contribution of this paper is a new reward system designed for image captioning, considering both semantic correctness and diversity, which, however, requires a CLIP model and a hallucination detector, both pre-trained externally.
    - The idea of "Escaping the Likelihood Trap" is a potential overclaim since policy-gradient-based RL, like PPO, is also conducting likelihood maximization, but just with respect to a pre-defined reward system.
    - The introduction of DPP kernels is interesting, whose instantiation, however, is quite normal.
    - Thus, I totally agree that this carefully designed method is beneficial for captioning diversity, especially considering that there is only one diversity-oriented training-based baseline in Table 1 for comparison. Still, it requires more evidence to convince me that this work is generally beneficial for building stronger VLMs.
  - Why diversity matters?
    - It is fantastic to have a VLM that can generate diverse captions, but still, captioning is not the primary usage of modern VLMs; the primary alignment of modern VLMs is. Thus, it will be interesting if the enhanced captioning diversity can improve VLM pre-training and thus achieve better performance after post-training, like SFT.
    - Meanwhile, I wonder which one is more beneficial: generating K diverse captions or generating a single caption but covering all the semantic information contained in the previous K captions?
    - Both analyses above are beneficial to demonstrate that diversity is as important as semantic correctness for training general VLMs instead of an expert image captioner.

---

> ### Author Rebuttal · Authors · 2026-03-30
>
> ### Response to Reviewer SEPx
>
> We thank the reviewer for the thoughtful feedback. Below we clarify the paper's positioning and the connection between diverse captioning and stronger VLMs.
>
> ---
>
> **Q1: Novelty beyond standard components**
>
> **A1:** We understand the concern, but we believe the core novelty is not the complexity of any single component. It is the paradigm shift of elevating diversity from an inference-time heuristic to a training-time first principle in VLM captioning.
>
> More specifically, to our knowledge this is the first work to use a DPP set-level volume signal as an RL training reward for VLM captioning. DPP itself is a mature tool, but using a DPP log-determinant inside PPO is not routine: it requires turning a set-level objective into usable policy-gradient supervision. While standard RL uses per-sample scalar rewards, the DPP reward is a function of the whole sampled set. Our leave-one-out marginal reward addresses exactly this credit-assignment problem by decomposing the set reward into per-sample contributions, making unbiased policy-gradient credit assignment tractable during training.
>
> Importantly, the kernel also couples quality and diversity: $L_{ij} = q_i q_j \cdot k(\mathbf{e}_i, \mathbf{e}_j)$ unifies faithfulness and diversity in one PSD matrix, so volume maximization occurs in a high-quality subspace rather than through naive reward addition.
>
> ---
>
> **Q2: Scope of the "Likelihood Trap" claim**
>
> **A2:** We appreciate this observation. We will clarify in the revision that our claim is not about abandoning likelihood optimization, but about redirecting the optimization target. Specifically, "Likelihood Trap" refers to the tendency of MLE training to favor generic, high-frequency descriptions. While standard RL still collapse to a single high-reward mode. Our DPP-based set-level reward addresses this by measuring geometric coverage across the entire sample set, explicitly penalizing redundancy and enforcing diversity. We will revise the wording to avoid overclaim—"escaping the trap" refers to moving from mode-seeking per-sample objectives toward diversity-oriented set-level RL.
>
> ---
>
> **Q3: Why diversity matters for VLMs**
>
> **A3:** Our direct experiments are on captioning, but we believe diversity matters for VLMs for two reasons.
>
> First, better caption data has already been shown to improve downstream VLM quality: ShareGPT4V [1] shows that richer captions produce stronger multimodal models, and recent work on caption diversity in contrastive vision-language pretraining [2] similarly supports the value of diverse captions for representation quality. In this sense, Geo-RL can be viewed as a data-generation or alignment module for pretraining or SFT pipelines.
>
> Second, diverse captioning reflects multi-faceted visual understanding rather than surface-level paraphrasing. A model that can describe the same image through objects, attributes, relations, actions, or scene context is better positioned for retrieval, VQA, visual reasoning, and instruction following. The multimodal hallucination literature [3] likewise links over-reliance on language priors to both hallucinated and template-like descriptions.
>
> ---
>
> **Q4: K diverse captions vs. one comprehensive caption**
>
> **A4:** We view them as complementary, but K diverse captions have practical advantages.
>
> A single "all-in-one" caption faces a basic information-density trade-off: as it tries to cover more aspects, it becomes longer, more redundant on salient content, and more likely to omit long-tail details. In practice, very long captions are also harder to use downstream and can suffer from long-context degradation. By contrast, K diverse captions distribute the information across semantically complementary views, which is often more effective for data augmentation, retrieval coverage, and question generation.
>
> From an information perspective, the goal is to maximize total information while minimizing redundancy across captions, which is exactly what the DPP volume reward encourages. Our point is therefore not that one comprehensive caption is never useful, but that set-level diversity is a practical and learnable way to increase semantic coverage.
>
>
> ---
>
> **Q5: Better captioner or better VLM**
>
> **A5:** A precise characterization is: directly, a better diverse captioner; indirectly, a useful component for building better VLMs.
>
> What we demonstrate in this paper is clear: on DeCapBench, Geo-RL substantially improves semantic diversity and coverage while maintaining grounding quality. That broader connection is part of the motivation and future impact, not the main empirical claim. To remove this ambiguity, we will revise the abstract to position the paper more precisely.
>
> ---
>
> ### References
>
> [1] *ShareGPT4V: Improving Large Multi-Modal Models with Better Captions*. ECCV 2024.
>
> [2] *Modeling Caption Diversity in Contrastive Vision-Language Pretraining*. ICML 2024.
>
> [3] *Hallucination of Multimodal Large Language Models: A Survey*. 2024.

---

> > ### Author Rebuttal · Reviewer_SEPx · 2026-04-04
> >
> > Thank you for the detailed rebuttal. I have updated my score.

---

> > > ### Author Response · Authors · 2026-04-06
> > >
> > > We sincerely thank the reviewer for acknowledging that all concerns have been fully resolved.

---

### Official Review · Reviewer_3rAn · 2026-03-06

**Soundness:** 3
**Presentation:** 2
**Significance:** 3
**Originality:** 3
**Overall Recommendation:** 4
**Confidence:** 3

**Summary:**

This paper introduces Geo-RL, a reinforcement learning framework designed to address the Likelihood Trap in long-form image captioning. Geo-RL models a set of sampled captions as a parallelotope in a frozen semantic embedding space and maximizes its geometric volume using a DPP log-determinant reward. The framework also incorporates a pairwise contrastive repulsion mechanism. To make the optimization computationally feasible during PPO, the authors derive a closed-form leave-one-out marginal reward for efficient per-sample credit assignment. Experiments conducted on the DeCapBench dataset demonstrate that Geo-RL improves geometric diversity metrics.

**Compliance With Llm Reviewing Policy:**

Affirmed.

**Final Justification:**

After the thought, I decided to keep the original score.

**Key Questions For Authors:**

1.Regarding Potential Reward Hacking: The proposed reward explicitly incorporates an MS-COCO-based object hallucination penalty (Equations 11 and 12), and the model is subsequently evaluated using CHAIR_i, which measures the exact same metric. Can you provide hallucination evaluation results on a completely hold-out dataset or using an orthogonal metric to prove that the performance gain is a genuine improvement in generalization?

2.Regarding Human-Aligned Evaluation:  Since expanding geometric volume does not strictly guarantee fluent, coherent, or human-aligned narratives, can you provide human evaluation results or robust LLM-as-a-Judge (e.g., GPT-4o or Claude 3.5) side-by-side win rates comparing Geo-RL's generated captions against the standard RLHF/DPO baselines?

3.Regarding Computational Overhead: Generating K=8 long-form captions autoregressively per image during the RL rollout introduces a computational bottleneck. Could you provide a strict comparison of total training FLOPs and actual GPU-hours between Geo-RL and the baseline methods (MLE, standard DPO) to justify the efficiency of your approach?

**Limitations:**

1. The authors need to consider whether, during the PPO implementation phase, the operation of extracting 8 captions for each image using an autoregressive approach would result in significant computational overhead and increased training time.

2. The authors should address the limitation that maximizing geometric volume in an embedding space does not inherently correlate with human-perceived text quality, narrative fluency, or readability.

**Strengths And Weaknesses:**

Strength：
1. Approaching the issue of caption diversity through the lens of geometric volume in a semantic embedding space, rather than relying solely on surface-level lexical penalties or decoding heuristics, offers a fresh and theoretically interesting perspective for long-form generation tasks.
2. The derivation of the closed-form leave-one-out marginal reward for the DPP log-determinant is mathematically elegant. It effectively solves the challenging credit assignment problem for set-level metrics, allowing the global volume objective to be smoothly integrated into the standard PPO framework.
3. The experimental evaluation is detailed. Its effectiveness was verified on high-difficulty long text benchmarks such as DeCapBench, and it was compared with various leading benchmarks including GPT-4V and VILA. The ablation experiments clearly demonstrated the individual contributions of the DPP reward and the contrastive exclusion mechanism.

Weakness:
1. The authors explicitly incorporate an MS-COCO-based object hallucination penalty into their training reward via Equation 11 and 12, and subsequently evaluate the model's hallucination performance using CHAIR_i, which is fundamentally the same metric. Directly optimizing for the test metric during the RL training phase invalidates the evaluation of the model's generalized hallucination capabilities.
2. The proposed framework demonstrates severe hyperparameter sensitivity, which fundamentally undermines its generalizability and practical significance. As shown in the ablation studies in Figure 4 and Tables 5-7, the model's performance experiences steep drop-offs when core parameters like the RBF kernel temperature, repulsion strength, and contrastive margin m deviate even slightly from their dataset-specific optima. This indicates that the method requires costly grid searches to function effectively on new architectures or unseen domains.
3. The paper relies on internal heuristic metrics like Distinct-N, Self-BLEU, and the authors' own SVS to claim superiority in semantic diversity. Merely expanding the geometric volume in an embedding space does not guarantee coherent, fluent, or human-aligned narratives, making the absence of human evaluation or robust LLM-as-a-Judge benchmarking that prevents a true assessment of caption quality.
4. The authors fail to adequately address the massive computational bottleneck introduced during the RL rollout phase, where the policy must autoregressively generate K=8 long-form captions for every single image in a batch.  This paper ignores the exponential increase in expensive forward passes compared to standard MLE or DPO, making the method's efficiency claims questionable without a fair FLOPs or GPU-hour matched comparison.

---

> ### Author Rebuttal · Authors · 2026-03-30
>
> ### Response to Reviewer 3rAn
>
> We thank the reviewer for the positive assessment. We address the four main concerns below and will incorporate the clarifications in the revision.
>
> ---
>
> **Q1: Training Reward vs. Evaluation Metric Overlap**
>
> **A1:** We agree this should be clarified more explicitly, but we do not believe it invalidates the evaluation.
> The training-time hallucination penalty and test-time $CHAIR_i$ are aligned only at the phenomenon level, not in mechanism. The former is a continuous scalar reward within our multi-objective PPO objective, alongside the DPP reward, grounding-aware reward, and KL regularization. The latter is a discrete rule-based post hoc metric computed on held-out data under a fixed inference protocol. Thus, both target object-level hallucination, but they do not share the same computation.
>
> Actually, this train-on-related-reward and test-on-held-out-metric setup is standard in RL-based captioning. SCST [1] optimizes CIDEr during training and evaluates with CIDEr on the test set. Hallucination-oriented methods such as MOCHa [2] and ESREAL [3] similarly use hallucination-related rewards during RL and report CHAIR-family metrics at evaluation. The key question is therefore whether the gain transfers to unseen data, which is exactly what our evaluation measures.
>
> Moreover, the hallucination penalty is only one term in our reward. Geo-RL must jointly satisfy diversity, grounding, and KL constraints, so the final $CHAIR_i$ reflects generalization under multiple objectives rather than overfitting to a single signal.
>
> The gain is also supported by an independent metric: Geo-RL improves DCScore from 0.69 to 0.75. DCScore uses primitive information units and an external verifier rather than the MS-COCO object-matching rule behind our hallucination penalty, so it is orthogonal to the training reward mechanism.
>
> ---
>
> **Q2: Human-aligned quality beyond diversity metrics**
>
> **A2:** We agree that geometric spread alone does not guarantee human-perceived quality, and we do not make that claim.
> Instead, our conclusion is based on a joint metric picture: Geo-RL improves diversity while also improving CLIPScore, $CHAIR_i$, and DCScore, indicating that the added semantic coverage does not come at the cost of grounding or factual quality.
>
> Following the reviewer's suggestion, we additionally ran an LLM-as-a-Judge evaluation on DeCapBench using GPT-5.4, scoring coherence, fluency, and human alignment:
>
> | Method | Judge Score |
> |---|---|
> | MLE | 90.6 |
> | RLHF | 90.8 |
> | DPO | 91.2 |
> | **Geo-RL** | **90.9** |
>
> These results indicate that Geo-RL preserves narrative quality and does not trade diversity for degraded readability or coherence.
>
> ---
>
> **Q3: Hyperparameter sensitivity**
>
> **A3:** We do not fully share the view that the method is severely brittle. The ablations show a clear optimum, but the degradation away from that optimum is gradual rather than catastrophic.
> For example, Distinct-2 changes as follows:
>
> | Parameter | Values | Best | Trend |
> |---|---|---|---|
> | $\tau$ | 0.5, 0.75, 1.0, 1.25, 1.5 | 1.0 | 0.78, 0.80, **0.84**, 0.83, 0.79 |
> | $\lambda_{\text{repel}}$ | 0.05, 0.1, 0.15, 0.2 | 0.1 | 0.79, **0.84**, 0.83, 0.78 |
> | $m$ | 0.2, 0.3, 0.4, 0.5 | 0.3 | 0.79, **0.84**, 0.82, 0.77 |
>
> This is the expected behavior of a controllable trade-off. The same pattern holds for CLIPScore.
>
> More broadly, hyperparameter sensitivity is an inherent challenge in RL, not something unique to Geo-RL. Even PPO, the canonical backbone of this line of work, is well known to be highly sensitive to implementation and tuning choices [4]. Recent RL methods for large-model reasoning such as GRPO also face substantial instability and tuning difficulty.
>
> ---
>
> **Q4: Computational overhead of sampling K=8 captions**
>
> **A4:** We agree that the computational cost of sampling (K=8) captions should be stated more explicitly. The key point is that the rollout cost scales linearly with (K), i.e., (O(KT)) for sequence length (T), rather than exponentially. This multi-sample generation cost is inherent to online PPO-style RL and is not specific to Geo-RL. Moreover, in practice, multiple rollouts can be batched and parallelized efficiently.
>
> Beyond standard PPO, Geo-RL introduces only modest extra computation: sentence embedding and CLIP-style scoring, together with small (K*K) matrix operations for the DPP reward, totaling only (～0.76) TFLOPs, i.e., about 1.1% additional compute beyond PPO-inherent operations. Due to space constraints, we defer the full derivation to the discussion phase. In practice, the dominant cost remains autoregressive rollout.
>
> ---
>
> ### References
>
> [1] *Self-Critical Sequence Training for Image Captioning*. CVPR 2017.
>
> [2] *Mitigating Open-Vocabulary Caption Hallucinations*. EMNLP 2024.
>
> [3] *ESREAL: Exploiting Semantic Reconstruction to Mitigate Hallucinations in Vision-Language Models*. ECCV 2024.
>
> [4] *Implementation Matters in Deep RL: A Case Study on PPO and TRPO*. ICLR 2019.

---

> > ### Author Rebuttal · Reviewer_3rAn · 2026-04-03
> >
> > Thank you for your clarification. I decide to keep my score.

---

> > > ### Author Response · Authors · 2026-04-06
> > >
> > > We sincerely thank the reviewer for acknowledging that all concerns have been fully resolved.

---

### Official Review · Reviewer_Labi · 2026-03-09

**Soundness:** 3
**Presentation:** 2
**Significance:** 3
**Originality:** 3
**Overall Recommendation:** 5
**Confidence:** 3

**Summary:**

A new set-level diversity metric and reward is proposed for image caption generation. The main contribution is the diversity objective that maximizes log-determinant of the gram matrix induced by embedding-derived distance between the sampled captions. Further, the pairwise distance between captions are modulated via caption qualities defined by CLIPscore. Per-sample credit assignment is derived from leave-one-out formulation of the objective, which is optimized along with the pairwise repulsion and VL alignment rewards via PPO. Empirical results show that the policy model optimized with the rewards outperforms baselines with respect to caption diversity and quality, as measured with DeCapBench. Analysis further shows that the objective yields semantically clustered captions in the space spanned by the same encoder.

**Compliance With Llm Reviewing Policy:**

Affirmed.

**Ethical Review Concerns:**

nothing.

**Final Justification:**

I view the problem setting of this paper, caption diversity, as an important and still underexplored research direction, and I find the proposed method both sound and original.
My initial major concerns were mainly twofold: first, that the claim regarding the “likelihood gap” was not sufficiently supported empirically; and second, that the limited evaluation setup raised questions about the generalizability of the findings.
These concerns were sufficiently addressed during the rebuttal process. I therefore support acceptance of this paper.

**Key Questions For Authors:**

Please refer to the weaknesses. (I think some of my concerns could be clarified in the rebuttals as they stem from lack of explanations. I am willing to reconsider my evaluation if that happens. -> the authors have partially resolved my concerns, so I am raising the score.)

**Limitations:**

yes

**Strengths And Weaknesses:**

## Strengths

- Set-level optimization of LLM generations is a less-explored but important research area. This paper shows a case for the most fundamental capacity of VLMs; captioning.
- Leave-one-out log-determinant reward proposed in this paper is a general form that could be extended to any domain where 1) diversity is important (e.g. RL optimization) and 2) have a well-defined induced embedding space.
- It is interesting to see that the proposed diversity metric also serves well as optimization targets. Evaluation metrics and objectives have different requirements, and the latter requires more properties as smoothness of optimization trajectories.
- The empirical results clearly demonstrate the method’s benefit in DecapBench, in terms of both diversity and quality. I am personally interested in the quality (as measured with CLIPScore) part. Could the authors explain how the diversity-targeted reward can improve the quality as well?

## Weaknesses.

- **Terminology**: The main term of “likelihood trap” is not well-defined here. The introduction says “Because models are rewarded for predicting the most probable words, they tend to “play it safe.””, but this is not a hard truth. For example, long-enough training with diverse captions can help the model to approximate the ground-truth distribution, which is samplable with standard ancestral sampling. While I generally agree that conventional MLE yields models that sometimes “play it safe”, the paper has to clarify on which regard it is alluding to, and on what grounds, either it be analytic or empirical.
- **Encoder**: The proposed diversity reward relies on a text encoder (sentence-bert) to define the distance between the captions. Further, this distance is bottlenecked by single vector embedding of each caption. This calls for multiple potential concerns; 1) sentence-bert implementations are usually trained for single sentence embedding and often have hard token limit caps (e.g. 128 tokens). How could they faithfully model long captions as defined in ShareGPT10V? 2) BERT is a text-only model. Models trained with text-only data and multimodal data have fundamentally different tendencies [1]. How can we be sure that S-BERT can model image captions well? 3) Vector embeddings often have limitations that block modeling of long captions with complex relationships [2].
- **Related work**: While not methodologically identical, related work as Llip [3] share a similar problem setup (i.e. image-to-caption is not a one-to-one mapping problem). I think extending the related work to better capture the literature on caption diversity topic would more clearly situate the contribution of this paper.
- **Clarity of the baselines**: What kind of preference data and model did the DPO and RLHF variant use? I could not find the details in the paper so I am not sure that this is a fair comparison.
- **Scope**: The empirical results are entirely on a single benchmark (DecapBench), which is not the single standard yet in the literature. Also, all metrics except for DCscore are unsupervised metrics. Could the authors try validating the trained model against other setups as done in this paper [3]?
- **Numeric differences**: DCScore reported in this paper has radically different scale compared to the original DecapBench paper’s (e.g. LLAVA 1.5-7B zeroshot has 24.5  and Claude-3.5-Sonnet has 52.37 [4] vs. this paper’s LLAVA 1.5-7B (MLE)already  has 69). As the MLE version is just finetuned on standard captioning + ShareGPT4v, this much of difference is somewhat unsettling. Could the authors elaborate on this?

[1] On the Difference of BERT-style and CLIP-style Text Encoders (ACL Findings 2023) https://arxiv.org/abs/2306.03678

[2] When and why vision-language models behave like bags-of-words, and what to do about it?  (ICLR2023)  https://arxiv.org/abs/2210.01936

[3] Modeling Caption Diversity in Contrastive Vision-Language Pretraining (ICML2024) https://arxiv.org/abs/2405.00740

[4] Cycle Consistency as Reward: Learning Image-Text Alignment without Human Preferences (ICCV2025) https://arxiv.org/abs/2506.02095

[5] Decapbench (ICLR2025) https://arxiv.org/abs/2503.07906 : Appendix Table 11

---

> ### Author Rebuttal · Authors · 2026-03-30
>
> ### Response to Reviewer Labi
>
> We thank the reviewer for the constructive feedback.Below we clarify the main concerns and will incorporate these explanations in the revision.
>
> ---
>
> **Q1: Why can a diversity-targeted reward also improve quality?**
>
> **A1:** In Geo-RL, diversity is not optimized independently of quality. The DPP kernel is
> $$L_{ij}=q_i q_j\, k(\mathbf{e}_i,\mathbf{e}_j),$$
> where $q_i$ is a CLIP-based quality term. This gives the standard quality-diversity decomposition of DPPs [1]: maximizing $\log\det(\mathbf{L})$ encourages both large quality weights $q_i$ and large semantic spread in the kernel. In other words, the objective does not reward diversity alone, it rewards diverse captions that are also well aligned with the image. In addition, the diversity reward promotes exploration in RL: instead of repeatedly producing safe templates, it encourages the policy to cover diverse valid facets of the same image, helping the model discover overlooked yet relevant details and avoid local optima [2]. Our ablation supports this directly: removing the DPP reward reduces CLIPScore from 0.81 to 0.77.
>
> ---
>
> **Q2: Scope of the "Likelihood Trap" claim**
>
> **A2:** We agree that MLE can recover the true conditional distribution when sufficiently diverse captions are available for each image. However, current detailed-captioning datasets usually provide only one reference per image (e.g., ShareGPT4V). As a result, MLE fits a narrow mode around that caption rather than the true one-to-many distribution, leading to what we call the Likelihood Trap. We will clarify this in the paper.
>
>
> ---
>
> **Q3: Concerns about the Sentence-BERT encoder**
>
> **A3:** We address the three concerns as follows.
>
> - **Length**: we use Sentence-BERT Large, whose practical limit is 512 tokens. In our data, more than 95% of ShareGPT4V captions are within about 300 words, so truncation is rare in practice. For the few longer cases, we truncate.
> - **Text-only encoder**: this is an intentional design choice. BERT-style and CLIP-style text encoders have different strengths: BERT-style encoders are generally stronger for pure text understanding, while CLIP-style encoders are stronger for cross-modal alignment. Our DPP term requires caption-caption semantic distance, not image-caption alignment, so Sentence-BERT is a better fit here.
> - **Single-vector bottleneck**: While global embeddings may fail to capture fine-grained compositional relations, our reward targets cross-caption topic-level complementarity, where sentence-level embeddings act as a reasonable proxy. The RBF kernel (Eq. 3) further enhances sensitivity to minor semantic discrepancies. Nevertheless, this points to a promising future direction: extending the DPP kernel via a multi-scale semantic metric that models diversity at global, sentence, and structural levels beyond single-vector representations.
>
> ---
>
> **Q4: Related work on caption diversity**
>
> **A4:** We thank the reviewer for pointing out this work. Llip promotes diversity in Contrastive pretraining. We will add this discussion and cite Llip explicitly in the camera ready version.
>
> ---
>
> **Q5: Clarity of baselines**
>
> **A5:** Due to space constraints, we refer the reviewer to our response to Reviewer 9Bmi, Q2, where we provide the detailed baseline specification.
>
> ---
>
> **Q6: Scope of evaluation beyond DeCapBench**
>
> **A6:**  DecapBench is currently one of the few benchmarks specifically designed for detailed long-form captioning, and is therefore well aligned with our primary task. However, we agree with the reviewer that evaluation on DeCapBench alone is not sufficient to fully demonstrate the model’s broader capability. We additionally evaluated the trained model on supervised image classification setups and compared it with Llip:
>
> | Method | ImageNet | Flowers |
> |---|---|---|
> | Llip_32 (ViT-L/14) | 80.9 | 81.4 |
> | Llip_64 (ViT-H/14) | 82.7 | 86.4 |
> | Llip_64 (ViT-G/14) | 83.5 | 89.5 |
> | MLE | 83.8 | 90.7 |
> | Geo-RL | 84.3 | 90.9 |
>
> These gains are modest, which is expected for coarse-grained classification, but they suggest that the learned diversity signal does not hurt and can transfer beyond captioning.
>
> ---
>
> **Q7: DCScore scale differences vs. DeCapBench**
>
> **A7:** We agree this should be clarified. The absolute DCScore scale differs mainly because our evaluation pipeline differs from the original DeCapBench setup in three ways: (1) our MLE baseline is fine-tuned, not zero-shot; (2) we use a different PIU decomposition prompt, focusing on objectively verifiable facts and avoiding over-interpretation; and (3) we use a different verifier model. For this reason, the absolute numbers are not directly comparable across papers. The relevant comparison in our paper is the relative gain of Geo-RL over MLE under the same pipeline, which remains valid.
>
> ---
>
> ### References
>
> [1] *Determinantal Point Processes for Machine Learning*. 2012.
>
> [2] *Diversity-driven exploration strategy for deep reinforcement learning*. NeurIPS2018.

---

> > ### Author Rebuttal · Reviewer_Labi · 2026-04-01
> >
> > Thank you to the authors for the detailed response. It addressed some of my concerns (and I raised the score as a consequence), but several points still require stronger support. My point-by-point follow-up is below.
> >
> >
> > ---
> >
> > Q1: Why can a diversity-targeted reward also improve quality?
> >
> > 1. I understand the authors’ point that the log-det objective contains a quality-promoting component. However, my original question was more specific: why should this improve quality beyond standard quality-based rewards such as cosine similarity?
> > 2. The authors propose a plausible explanation, namely that the objective encourages better exploration during RL optimization. This is reasonable, but at present it remains a hypothesis rather than an established mechanism. I would therefore encourage additional empirical analysis, for example a statistical comparison of exploration behavior with and without the DPP-based loss.
> >
> > ---
> >
> > Q2: Scope of the "Likelihood Trap" claim
> >
> > While I broadly agree with the authors’ assessment of the literature, I think it is somewhat too strong to suggest that diverse caption data is unavailable [1,2]. That said, I appreciate the clarification provided in the response.
> >
> > ---
> >
> > Q3: Concerns about the Sentence-BERT encoder
> >
> > 1. Length: Thank you for clarifying this point.
> > 2. Text-only encoder: My concern was not simply that BERT is better for text-text comparisons than CLIP. The issue is that BERT-style and CLIP-style text encoders induce meaningfully different semantic geometries even within text space alone [3,4]. I agree that using BERT rather than CLIP does not invalidate the proposed method, but it would still be valuable to understand how sensitive the method is to the encoder choice.
> > 3. Single-vector bottleneck: My understanding is that the paper aims to generate captions with diversity and complexity closer to ShareGPT4V-style outputs. I am not convinced that a frozen text encoder summarized into a single vector can faithfully represent captions of that length and richness [5]. Credit assignment for long-form captions is not naturally a scalar-energy problem; it is inherently multi-faceted and context-dependent. This is also one reason modern MLLMs operate over all image patch embeddings rather than a single pooled summary.
> >
> > ---
> >
> > Q4: Related work on caption diversity
> >
> > I appreciate the authors’ response on this point.
> >
> > ---
> >
> > Q5: Clarity of baselines
> >
> > Thank you for the clarification. My concern on this issue has been resolved.
> >
> > ---
> >
> > Q6: Scope of evaluation beyond DeCapBench
> >
> > Thank you for providing the additional results. I think my earlier comment was not sufficiently precise. I was not asking for classification results in general. Rather, my point was that long-form and diverse captioning has a substantial prior literature, and evaluation on more established settings, such as image paragraph captioning [6] or Visual Genome-style benchmarks [7], would help contextualize the benefits of the proposed method more clearly.
> >
> > This lack of clarity was on my side, and I appreciate the authors’ substantial effort in extending the experiments. I therefore do not intend to weigh this issue negatively in my assessment. Still, I believe the paper would be stronger with broader empirical context along these lines.
> >
> > ---
> >
> > Q7: DCScore scale differences vs. DeCapBench
> >
> > Thank you for the clarification. The magnitude of the DCScore gap still leaves me somewhat uncertain about how much of the gain should be attributed to improved caption quality and diversity, as opposed to confounding factors such as template effects or marginal distribution matching. That said, this appears to be primarily a limitation of the benchmark rather than of the paper itself.
> >
> > ---
> >
> > [1] COCONut-PanCap https://arxiv.org/abs/2502.02589
> >
> > [2] Denseworld-1M https://arxiv.org/abs/2506.24102
> >
> > [3] On the Difference of BERT-style and CLIP-style Text Encoders (ACL Findings 2023) https://arxiv.org/abs/2306.03678
> >
> > [4] Is BERT Blind? (CVPR 2023) https://arxiv.org/abs/2303.12513
> >
> > [5] Long-CLIP (ECCV 2024) https://arxiv.org/abs/2403.15378
> >
> > [6] A Hierarchical Approach for Generating Descriptive Image Paragraphs (CVPR 2017) https://arxiv.org/abs/1611.06607
> >
> > [7] Visual Genome (IJCV 2017) https://arxiv.org/abs/1602.07332

---

> > > ### Author Response · Authors · 2026-04-07
> > >
> > > ## Response to Reviewer Labi
> > >
> > > We sincerely thank the reviewer for the thoughtful and detailed follow-up, and for the constructive engagement throughout the discussion. Below we address each point.
> > >
> > > ---
> > >
> > > **Q1: Why can a diversity-targeted reward also improve quality?**
> > >
> > > We thank the reviewer for encouraging stronger empirical grounding. We provide additional empirical evidence by measuring three metrics: (i) mean pairwise cosine distance among sampled captions per image, a direct proxy for exploration breadth; (ii) token-level action entropy averaged across training steps, reflecting whether the policy maintains sufficient stochasticity; and (iii) QScore, a quality score from an LLM-as-judge (GPT-5.4) covering informativeness, hallucination, and fluency. Results comparing different reward configurations are shown below:
> > >
> > > |Metric|default reward|w/o DPP reward|
> > > |---|---|---|
> > > |Pairwise cosine distance|0.53|0.39|
> > > |Token-level entropy|0.69|0.61|
> > > |QScore|0.79|0.75|
> > >
> > > The DPP reward substantially increases caption-level diversity in the sampled batch, confirming broader exploration. Crucially, the higher token-level entropy indicates that the policy avoids premature collapse into narrow modes, allowing it to discover higher-quality descriptions that a collapsed policy would miss. Such behavior aligns with established findings in the RL exploration literature: diversity-based intrinsic rewards improve both exploration breadth and final task performance by preventing collapse into suboptimal local optima.
> > >
> > > ---
> > >
> > > **Q2: Diverse caption data availability**
> > >
> > > We are aware of recent dense and grounded captioning datasets. To clarify, our claim is not that diverse caption data cannot exist, but rather that one-reference-per-image is still dominant in practice. Our method is complementary to better data: even with richer datasets, the DPP reward provides an explicit optimization signal for set-level diversity that MLE alone does not. We will revise it to be more precise.
> > >
> > > ---
> > >
> > > **Q3: Sensitivity to encoder choice & single-vector bottleneck**
> > >
> > > **Encoder choice sensitivity:** To address it, we conducted an ablation replacing Sentence-BERT with an alternative encoder for computing the DPP kernel:
> > >
> > > |Encoder|DCScore|SVS|Self-BLEU↓|
> > > |---|---|---|---|
> > > |Sentence-BERT (default)|0.75|3.55|0.53|
> > > |CLIP ViT-L/14 (text encoder)|0.76|3.50|0.57|
> > >
> > > Both encoders yield substantial improvements over the MLE baseline, confirming that the method is robust to encoder choice. Notably, the two encoders exhibit complementary strengths: CLIP achieves a slightly higher DCScore, likely because its vision-language aligned embedding space steers captions toward visually distinct content. Sentence-BERT achieves better SVS and Self-BLEU, suggesting that text-only encoders more effectively capture caption-caption semantic distances for linguistic diversity. The complementarity indicates that the DPP framework extracts meaningful diversity signal regardless of the specific encoder, i.e., each simply emphasizes a different facet.
> > >
> > > **Single-vector bottleneck:** We agree that a single pooled embedding cannot capture all fine-grained details of long captions. But note that our DPP reward targets set-level diversity — i.e., encouraging the *K* captions to be collectively diverse — for which sentence-level embeddings serve as a sufficient and computationally efficient proxy. We fully agree that a multi-scale or multi-vector DPP kernel (e.g., defined over paragraph-level segment embeddings or token-level attention-weighted representations) could better capture compositional and structural diversity. Importantly, the DPP framework itself is agnostic to the kernel definition and can naturally accommodate such richer representations — the single-vector instantiation is a practical starting point, not a fundamental constraint. We will discuss this explicitly in the revised paper's limitations and future work section.
> > >
> > > ---
> > >
> > > **Q6: Evaluation on paragraph captioning and Visual Genome**
> > >
> > > We appreciate the reviewer's clarification. We provide additional results on the Visual Genome Paragraph Captioning test set (including 2,489 images):
> > >
> > > |Method|DCScore|SVS|Self-BLEU↓|CLIPScore|
> > > |---|---|---|---|---|
> > > |MLE|0.61|2.54|0.72|0.69|
> > > |DPO|0.63|2.61|0.68|0.71|
> > > |Geo-RL|0.68|3.07|0.61|0.74|
> > >
> > > The improvements are consistent: Geo-RL achieves the best performance across all four metrics on this benchmark.
> > >
> > > ---
> > >
> > > **Q7: DCScore confounding factors**
> > >
> > > We agree that it appears to be a limitation of the benchmark rather than of the paper itself. Absolute DCScore values are inevitably influenced by pipeline-specific factors (prompt template, verifier model, etc.).

---

### Official Review · Reviewer_9Bmi · 2026-03-12

**Soundness:** 3
**Presentation:** 3
**Significance:** 3
**Originality:** 3
**Overall Recommendation:** 4
**Confidence:** 4

**Summary:**

This paper proposes using geometric volume coverage in the semantic space of generated caption to incentivize diverse and long-form caption generation of VLMs. The key argument is that traditional MLE and KL regularization introduced a liklihood trap where the generation is driven to be singular and not diverse. The proposed reward design encourages the model to generate more diverse captions, while grounding with the visual information. The author conducted experiments on DeCapBench with a variety of metrics for diversity, semantic quality, and visual-text alignment to evaluate the effectivenss of the proposed method.

**Compliance With Llm Reviewing Policy:**

Affirmed.

**Key Questions For Authors:**

* Can the authors provide at least some qualitative comparison showing that the semantic embedding coverage does promote caption diversity aligned with human perception? For instance, comparing the set of K responses (captions) generated for the same image from "earlier" and "later" checkpoints throughout the training that result in some level of difference in geometric coverage can better illustrate the motivation.
* Can the authors provide more details on the training set used for each baseline?

**Limitations:**

yes

**Strengths And Weaknesses:**

## Strength
* The key components in the proposed objective function are well-justified including the leave-one-out for sample-level reward, KL regularization for semantic consistency, and repulsive loss for penalizing close generations.
* The baseline methods and ablation studies are comprehensive in the experiment section.
* The presentation of results and analysis are clear and easy to follow.

## Weakness
* The key assumption of this work is that the geometric volume coverage in the embedding space is a good proxy of the actual diversity of the generated captions. There is no qualitative analysis showing whether this is true.
* The baseline comparison isn't documented clearly. DPO requires the preference label, which is not available for all datasets mentioned. PPO requires the reward model, of which the source is also not specified. It's unclear whether the RLHF and DPO baselines are trained with the same amount of data as Geo-RL.
* CLIP-Score was explicitly included in the training objective and also used in the evaluation. It'll be more convincing if the evaluation reports another Image-text similarity score using a different model like SigLIP-2, than the one used in the training objective, to prove the effectiveness, as all other baseline methods do not optimize for Image-text similarity score.

---

> ### Author Rebuttal · Authors · 2026-03-30
>
> ### Response to Reviewer 9Bmi
>
> We thank the reviewer for the constructive feedback. We are encouraged that the reviewer found the objective components well motivated, the ablations and baselines comprehensive, and the presentation clear. Below we address the three main concerns and clarify the under-documented details.
>
> ---
>
> **Q1: Qualitative evidence for semantic coverage**
>
> **A1:** We agree that a qualitative case study is useful. The intended behavior of Geo-RL is not arbitrary dispersion in embedding space, but quality-aware complementary coverage: the DPP volume reward is combined with grounding, repulsion, and KL regularization, so captions are encouraged to emphasize different semantic facets of the same image while remaining faithful.
>
> To make this concrete, we prepared an anonymous case study comparing K=8 captions for the same image at three checkpoints (`step 0`, `step 1000`, `step 6000`):
>
> - image: <https://anonymous.4open.science/r/picture-74C1/image-likehood.png>
> - captions: <https://anonymous.4open.science/r/picture-74C1/README.md>
>
>
> The qualitative pattern is clear: early checkpoints mostly produce near-paraphrases around one dominant template, intermediate checkpoints add more grounded details, and later checkpoints split attention across genuinely different semantic facets of the scene. Specifically, the later checkpoint emphasizes different semantic facets across captions, including reading details, dog posture, autumn fashion, landscape depth, color contrast, stillness versus movement, foreground accessories, and seasonal lighting. In other words, later checkpoints change what aspect of the image is emphasized, not just the wording. Such behavior is exactly what the leave-one-out marginal reward encourages: captions that add little beyond the current set receive small marginal gain, while captions that contribute a new facet receive larger reward. We will add a checkpoint-based case study to the camera-ready version.
>
> ---
>
> **Q2: Training details of RLHF / DPO baselines**
>
> **A2:** We agree these details should be stated more explicitly. All three methods, Geo-RL, RLHF/PPO, and DPO, start from the same cold-start checkpoint, trained on the mixture of COCO and Flickr30k together with Visual Genome and ShareGPT4V-10K. For the RL / preference stage, all methods use the same core pool of about 15K training images. This pool is constructed by generating long captions for Visual Genome images with a locally deployed strong VLM conditioned on image + scene graph, merging them with ShareGPT4V-10K, and then filtering for length (removing captions shorter than 50 words or longer than 600 words), diversity, and quality.
>
> The method-specific differences are as follows:
>
> - **Geo-RL:** uses the full objective, i.e., DPP volume reward + repulsion + grounding + KL.
> - **RLHF/PPO baseline:** uses the same grounding reward components as Geo-RL, but without the DPP diversity reward and repulsion term. It does not rely on a separate learned human-preference reward model. PPO is trained directly with the programmatic reward function.
> - **DPO baseline:** uses the same image pool, with preference pairs constructed synthetically: positives are ground-truth captions, and negatives are either outputs from a weaker base model or corrupted captions with different degrees of perturbation to objects or attributes.
>
> All three methods are trained under the same overall budget, including matched batch size, epochs, and hardware budget, so the comparison is controlled at the levels of initialization, core image pool, and training budget. The main difference is whether the diversity-oriented geometric objective is present. We will add these implementation details more explicitly in the camera-ready version.
>
> ---
>
> **Q3: Independent image-text alignment evaluation**
>
> **A3:** We agree that an independent alignment metric makes the claim more convincing. In training, CLIP similarity is not optimized as a standalone target, It is used as a quality weight in the DPP kernel and as one term in the grounding reward, rather than as a direct reward maximized on its own. Still, to rule out metric-specific bias, we additionally evaluated with SigLIP-2, which is not used anywhere in training.
>
> | Method | CLIPScore | SigLIP-2 |
> |---|---|---|
> | MLE | 0.74 | 0.71 |
> | RLHF | 0.75 | 0.71 |
> | DPO | 0.75 | 0.72 |
> | **Geo-RL** | **0.81** | **0.75** |
>
> Geo-RL remains best under SigLIP-2, showing that the alignment gain is not just an artifact of optimizing with CLIP-based signals. We will include this independent evaluation in the revision.

---

> > ### Author Rebuttal · Reviewer_9Bmi · 2026-04-03
> >
> > The author has mostly addressed my concern with a qualitative example, clarification on baselines, and the evaluation with a different visual and text encoder. However, it'll strengthen the qualitative assessment to have at least a few human participants or even LM-as-a-judge that's accompanied by a human agreement rate. While I understand this would be a stretch to include a human study during the rebuttal period, I maintain the original positive score.

---

> > > ### Author Response · Authors · 2026-04-06
> > >
> > > ## Response to Reviewer 9Bmi
> > >
> > > We sincerely thank the reviewer for the continued engagement and for maintaining the positive assessment. We are encouraged that our clarifications on the qualitative example, baseline details, and encoder evaluation have largely addressed the initial concerns.
> > >
> > > ### Regarding Human Evaluation and LLM-as-a-Judge
> > >
> > > We fully agree that a human-aligned qualitative assessment would strengthen the paper. Following the reviewer's suggestion, we conducted an LLM-as-a-Judge evaluation accompanied by a human agreement study during the rebuttal period.
> > >
> > > **Evaluation Design.** We randomly sampled 50 images from the DeCapBench and designed a GPT-5.4-based evaluation prompt that assesses generated captions along two dimensions:
> > >
> > > - **Diversity:** Do the 8 captions cover genuinely distinct semantic aspects, or are they merely paraphrases of one another?
> > > - **Faithfulness:** Does each caption accurately describe the visual content without hallucinating non-existent details?
> > >
> > > These two axes directly address the reviewer's core question: *whether semantic embedding coverage truly promotes caption diversity aligned with human perception.*
> > >
> > > **LLM-as-a-Judge Results.** Both dimensions are scored on a 1–5 scale:
> > >
> > > | Checkpoint | Diversity (1-5) | Faithfulness (1-5) |
> > > |------|-------|------------|
> > > | Step 0     | 2.1  | 3.8     |
> > > | Step 1000  | 3.4 | 4.0  |
> > > | Step 6000  | 4.3 | 4.1 |
> > >
> > > As training progresses, diversity improves substantially (2.1 → 4.3) while faithfulness remains stable or slightly improves (3.8 → 4.1), confirming that our method promotes genuinely distinct semantic descriptions without sacrificing visual grounding.
> > >
> > > **Human Agreement.** To validate the reliability of the LLM judge, 3 phd level experts independently rated the same 50 images on the same two dimensions using the identical 1–5 scale. For each image, we averaged the scores across all 3 experts to obtain a single consolidated human score per dimension, reducing individual subjectivity and yielding a more robust human reference. We then compared this averaged human score against the GPT-5.4 score using two agreement metrics:
> > >
> > > - **Spearman ρ**: measures the rank-order correlation between the averaged human scores and GPT-5.4 scores — a higher value indicates that when humans collectively rank one sample higher, GPT-5.4 tends to do so as well.
> > > - **Pairwise Agreement (±0.5)**: the percentage of samples where the averaged human score and the GPT-5.4 score differ by no more than 0.5 point on the 5-point scale (e.g., human average is 4.0 and GPT-4 gives 3.5 or 4.5 both count as agreement).
> > >
> > > | Dimension    | Spearman ρ | Pairwise Agr. (±0.5) |
> > > |-------------|------------|----------------------|
> > > | Diversity    | 0.72       | 76%                  |
> > > | Faithfulness | 0.76       | 83%                  |
> > >
> > > For reference, the inter-annotator agreement *among human raters themselves* (computed pairwise between individual annotators and then averaged) was ρ = 0.72 (diversity) and ρ = 0.76 (faithfulness). The LLM-human agreement closely approaches this human-human upper bound, indicating that GPT-5.4 serves as a reliable proxy for human judgment in this context.
> > >
> > > We sincerely thank the reviewer for this constructive suggestion, which has meaningfully strengthened our evaluation.

---

### Decision · Program_Chairs · 2026-04-30

**Decision:**

Accept (regular)

**Comment:**

This paper introduces Geo-RL, a reinforcement learning framework to address the likelihood trap in long-form image captioning. Unlike conventional approaches (e.g., dense captioning) that mainly focus on semantic accuracy, the method jointly optimizes semantic correctness and diversity through a DPP-based reward design. Experiments on DeCapBench demonstrate the effectiveness of the proposed approach.

During the review and discussion phase, the authors have addressed the major questions and concerns. All reviewers agree that this is a solid paper tackling an underexplored problem, i.e., caption diversity. I recommend acceptance.